# Enriched conditioning expands the regenerative ability of sensory neurons after spinal cord injury via neuronal intrinsic redox signaling

Francesco De Virgiliis[1,6], Thomas H. Hutson [1,6], Ilaria Palmisano [1], Sarah Amachree[1], Jian Miao[1], Luming Zhou[1], Rositsa Todorova[1], Richard Thompson[2], Matt C. Danzi [3], Vance P. Lemmon [3], John L. Bixby [3], Ilka Wittig [4], Ajay M. Shah[2] & Simone Di Giovanni [1,5 ✉]

Overcoming the restricted axonal regenerative ability that limits functional repair following a central nervous system injury remains a challenge. Here we report a regenerative paradigm that we call enriched conditioning, which combines environmental enrichment (EE) followed by a conditioning sciatic nerve axotomy that precedes a spinal cord injury (SCI). Enriched conditioning significantly increases the regenerative ability of dorsal root ganglia (DRG) sensory neurons compared to EE or a conditioning injury alone, propelling axon growth well beyond the spinal injury site. Mechanistically, we established that enriched conditioning relies on the unique neuronal intrinsic signaling axis PKC-STAT3-NADPH oxidase 2 (NOX2), enhancing redox signaling as shown by redox proteomics in DRG. Finally, NOX2 conditional deletion or overexpression respectively blocked or phenocopied enriched conditioning-dependent axon regeneration after SCI leading to improved functional recovery. These studies provide a paradigm that drives the regenerative ability of sensory neurons offering a potential redox-dependent regenerative model for mechanistic and therapeutic discoveries.

---

[1] Division of Neuroscience, Department of Brain Sciences, Imperial College London, London, UK. [2] British Heart Foundation Centre, School of Cardiovascular Medicine and Sciences, James Black Centre, King's College London, London, UK. [3] Miami Project to Cure Paralysis, Center for Computational Sciences, University of Miami, Miami, FL 33136, USA. [4] Functional Proteomics, Faculty of Medicine, Goethe University, Frankfurt, Germany. [5] Laboratory for NeuroRegeneration and Repair, Center for Neurology, Hertie Institute for Clinical Brain Research, University of Tuebingen, Tuebingen, Germany. [6] These authors contributed equally: Francesco De Virgiliis, Thomas H. Hutson. ✉email: s.di-giovanni@imperial.ac.uk

Axon regeneration is severely limited following a spinal cord injury (SCI) in the adult mammalian central nervous system (CNS) leading to permanent impairment of sensory and motor function[1,2]. Previous approaches used to identify molecular pathways and targets involved in axon regeneration include the study of (i) developmental pathways[3–8], (ii) molecular screening in invertebrates and non-mammalian vertebrates[9–12], and (iii) injury-dependent mechanisms including the well-established conditioning lesion paradigm[13–17].

Axon regeneration after a conditioning lesion is elegantly displayed in sensory dorsal root ganglia (DRG) neurons. The pseudounipolar anatomy of DRG neurons allows them to convey afferent information from the periphery to the spinal cord and supraspinal structures. The regenerative potential of DRG neurons after a peripheral nerve injury or a SCI is enhanced by a prior injury of the peripheral axon: this phenomenon is known as the conditioning lesion. It was first characterized in the sciatic nerve over forty years ago[18] and in the spinal cord in the 80s and 90s[19,20]. Ever since, the conditioning lesion paradigm has been the gold standard for sensory neuron regeneration and led to the discovery of multiple regenerative signaling pathways[21–24]. However, the extent of axon regeneration elicited by the conditioning lesion is limited, especially in the spinal cord, where primed sensory neurons are still unable to regenerate large numbers of axons beyond the injury site[19].

Recent pre-clinical and clinical studies have shown that increasing the activity of sensory afferent axons following SCI is critical to sensory and motor recovery[25–29]. Therefore, extending the regenerative potential of these axons might be an important step towards improved functional recovery. We recently found that injury-independent approaches can also prime and enhance the regenerative potential of injured neurons. Exposing rodents to environmental enrichment (EE) rather than standard housing (SH) before axonal injury enhances the regenerative ability of DRG neurons via activity-mediated CREB-binding protein-dependent histone acetylation, which increases the expression of genes associated with the regenerative program[30]. This increase was comparable to the one observed after performing a conditioning sciatic nerve axotomy (SNA) prior to the SCI[30]. Therefore, both injury-dependent and -independent mechanisms can prime sensory neurons, leading to an increased axon regeneration after a SCI. However, it is not known whether their combination might be additive or synergistic, further propelling axon regeneration.

We developed a superior regenerative paradigm by combining EE with SNA (conditioning lesion) before a SCI. Coupling EE and SNA elicited an additive effect to the regenerative potential of sensory DRG neurons, significantly enhancing axon regeneration well beyond the spinal injury site. This regenerative model, which we call enriched conditioning, exhibits superior regenerative ability compared to EE or conditioning SNA alone. Mechanistically, we found that protein kinase c (PKC)-dependent phosphorylation of the transcription factor (TF) signal transducer and activator of transcription 3 (STAT3) induces its binding on hyperacetylated regions of the NADPH oxidase 2 (NOX2) promoters. This triggers the expression of the NOX2 complex enhancing redox signaling. NOX2 deletion in DRG neurons in vivo demonstrated that intrinsic NOX2 expression is required for enriched conditioning-dependent redox signaling and axonal regeneration. Lastly, NOX2 overexpression in DRG neurons promotes regeneration of sensory axons and functional recovery after SCI.

## Results

### Combining EE with SNA (enriched conditioning) primes DRG neurons for regeneration after SCI. We first examined whether

the SNA and EE-dependent regenerative ability of DRG sensory axons would implicate similar or distinct regenerative mechanisms. Comparing the gene expression profiles in DRG following pre-exposure to EE or SNA based upon a previously published[30] and a recently generated dataset, we found that the two conditions generated remarkably different sets of differentially expressed (DE) genes (P-value < 0.05) (Supplementary Fig. 1A and Supplementary Data 1) and Gene Ontology (GO) pathways (Supplementary Fig. 1B and Supplementary Data 2 and 3) (P-value < 0.05). Therefore, we hypothesized that combining the two conditions (EE + SNA) would result in an additive regenerative effect. Mice were placed in EE for 10 days before undergoing an SNA; 24 h after the SNA, the sciatic DRG were dissected and cultured. After 12 h in culture, we observed a twofold increase of DRG neurite outgrowth compared to EE or SNA alone (Fig. 1a, b). Furthermore, when we combined EE and SNA, and then performed a spinal cord T9 dorsal hemisection, we observed a significant increase in cholera toxin subunit B (CTB) traced sensory axons beyond the lesion site compared to SH sham, EE sham, or SH SNA (Fig. 1c–f). EE SNA did not affect the intensity of glial fibrillary acidic protein (GFAP) staining, a marker of the astrocytic scar that surrounds the lesion (Fig. 1d, g). Together, these data show that combining EE + SNA results in an additive effect that we call enriched conditioning, significantly increasing the regenerative ability of DRG neurons beyond the borders of a spinal cord lesion.

### Enriched conditioning upregulates the NOX2 complex in DRG neurons. To elucidate the molecular mechanisms underlying the enriched conditioning (EE + SNA)-dependent regenerative response, we performed RNA sequencing (RNAseq) from sciatic DRG following SH SNA and EE + SNA, and compared the expression profiles to control animals (SH Sham) and to a previously described dataset (EE Sham)[30]. We found 4023 DE genes (P-value < 0.05) following SH SNA, 789 following EE Sham, and 4601 following EE + SNA. Interestingly, EE + SNA and SH SNA have ~45% of DE genes in common, whereas EE + SNA has only a small percentage (~5%) of DE genes in common with EE Sham and only ~3% of the DE genes are shared between all the conditions (Fig. 2a). About 30% of DE transcripts were exclusively found following EE + SNA (Fig. 2a). To gain insight into the molecular pathways specifically modulated by EE + SNA, we performed GO (molecular functions (MFs), P-value < 0.05) functional analysis of DE genes across each experimental group (Supplementary Data 2–4). Although the GO analysis showed a degree of similarity between EE + SNA and SH SNA, we observed selected signaling pathways specifically associated with EE + SNA, including NADPH (NOX)-dependent redox signaling and phosphatase activity (red box Fig. 2b). A number of pathways were also significantly regulated after SNA or EE + SNA. However, GTPase activity and DAG-PKC signaling were significantly upregulated in EE + SNA compared to EE sham or SNA alone (yellow box Fig. 2b). Interestingly, NOX-dependent oxidation regulates and is in turn regulated by phosphatase activity[15,31–33]. Accordingly, a protein–protein interaction network using the NOX, phosphatase, GTPase, and DAG-PKC signaling pathways shows that they are functionally connected (Fig. 2c) and significantly enriched for interactions compared to random networks (Supplementary Data 5). Based on these findings, we hypothesized that NOX signaling could play a central role in the enriched conditioning (EE + SNA)-dependent increase in regenerative potential.

When we analyzed the genes represented in the NOX-dependent redox signaling GO cluster, surprisingly we found that all the NOX2 complex subunits were upregulated (P-value <

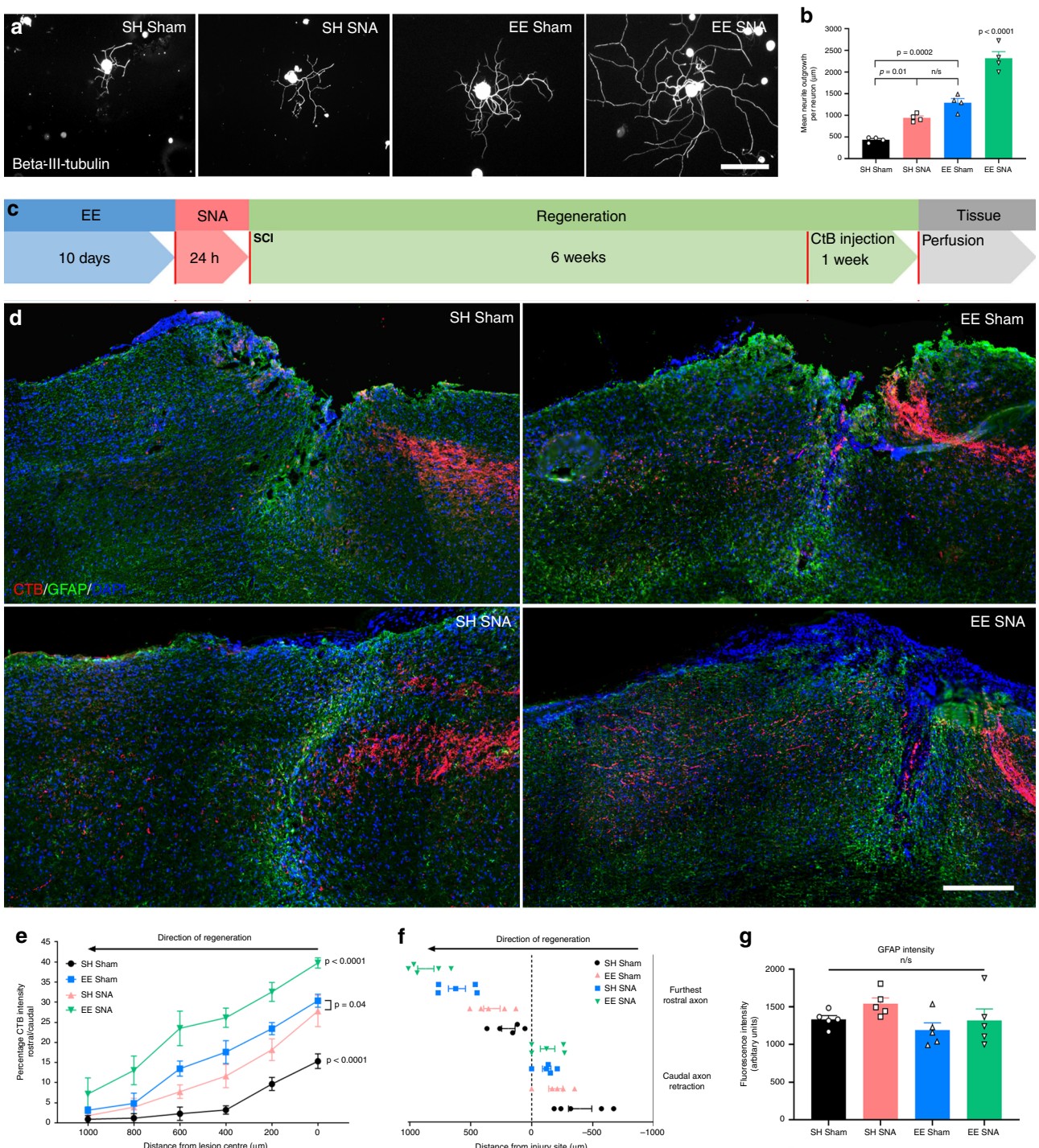

**Fig. 1 Enriched conditioning (EE + SNA) induces a significant increase in DRG regenerative growth in vitro and axon regeneration in vivo after SCI.**
**a** Representative images of cultured DRG neurons from SH sham, EE Sham, SH SNA, and EE + SNA stained for Beta-III-tubulin. Scale bar, 100 μm.
**b** Quantification of average neurite outgrowth per neuron (mean ± SEM, one-way ANOVA, Tukey's post hoc, n.s. = nonsignificant, $n = 4$ biologically independent animals/group, average of 20 cells/replicate). **c** Timeline for the in vivo experiment. **d** Representative images of CTB-traced (magenta) dorsal column sensory axons after injury, glial fibrillary acidic protein (GFAP) (green) and DAPI (blue). Scale bar, 200 μm. **e** Quantification of CTB-positive regenerating axons (mean ± SEM, Two-way repeated-measures ANOVA, Tukey's post hoc, $n = 5$ biologically independent animals/group. Fluorescence intensity was measured in one series of tissue for each spinal cord). **f** Average distance between lesion center and furthest regenerating axon/caudal tract retraction (mean ± SEM, $n = 5$ biologically independent animals/group. Length of the furthest regenerating five axons and main tract retraction was measured in one series of tissue for each spinal cord). **g** Quantification of GFAP intensity around the lesion site (mean ± SEM, one-way ANOVA, n.s. = nonsignificant, $n = 5$ biologically independent animals/group). Fluorescence intensity was measured in one series of tissue for each spinal cord.

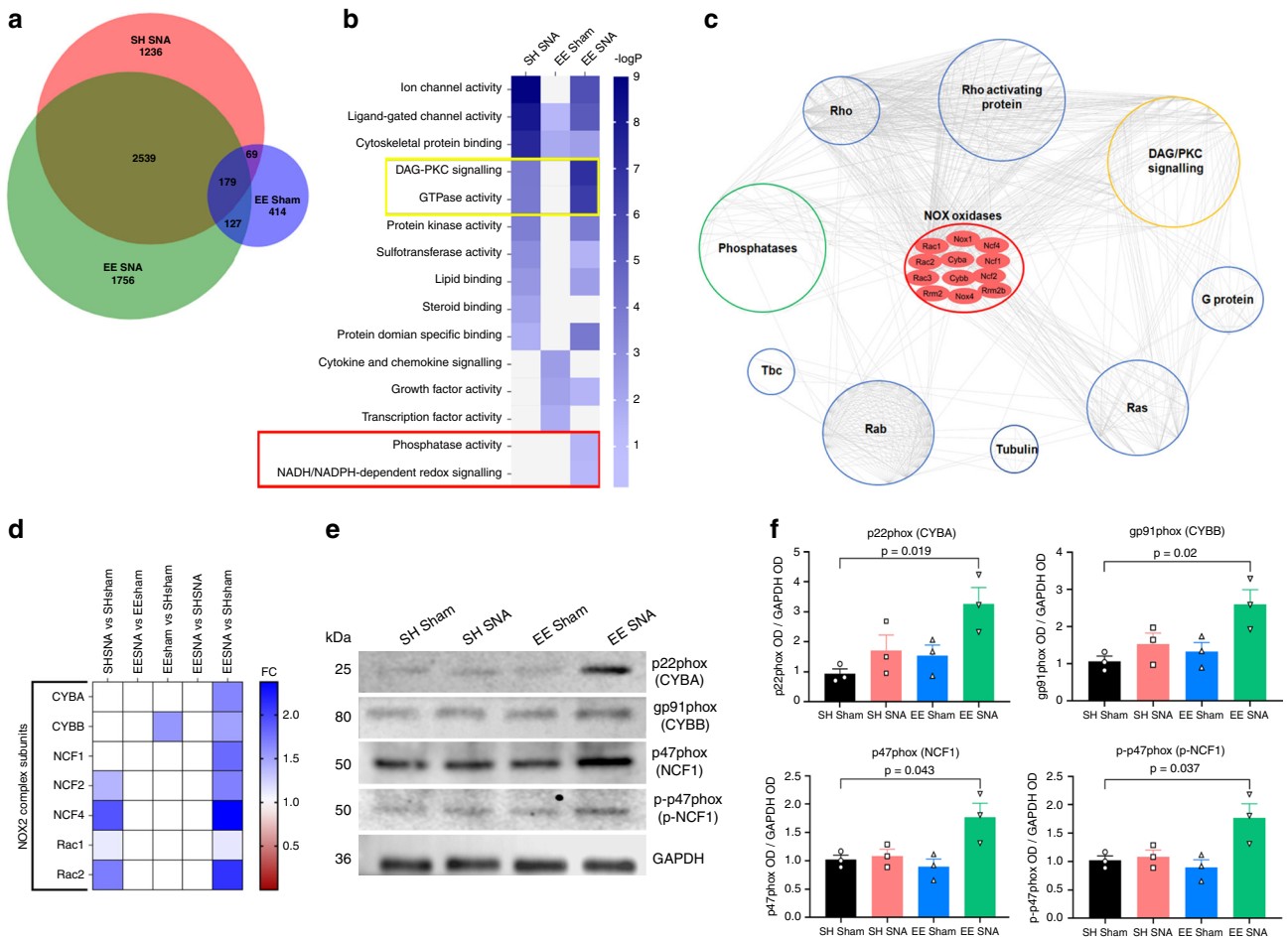

**Fig. 2 EE + SNA induces upregulation of signaling pathways and of NOX2 complex in DRG. a** Area proportional Venn diagram showing the number of differentially expressed (DE) genes and the extent of overlap among the three conditions SH SNA, EE Sham, and EE + SNA (each normalized to respective SH sham, *P*-value < 0.05, *n* = 3 independent biological experiments/condition). **b** Heatmap showing the Gene ontology analysis (molecular function, DAVID) of the DE genes modulated by SH SNA, EE Sham, and EE + SNA vs. SH Sham (modified Fisher's exact *P*-value < 0.05). **c** Network visualization of protein–protein interaction of the NOX oxidases category with proteins belonging to signaling pathways specifically enriched upon EE + SNA. Each protein is organized in a circular layout accordingly to their shared molecular function. Nox2 subunits are organized in the center of the network. Each line (edge) represents a protein–protein interaction. **d** Heatmap showing mRNA expression analysis (fold change vs. SH Sham, *P*-value < 0.05) of the genes belonging to NOX2 complex. **e** Immunoblotting analysis of NOX2 complex components from DRG protein extracts after SH Sham, EE Sham, SH SNA, or EE + SNA. **f** Quantification of immunoblotting, glyceraldehyde phosphate dehydrogenase (GAPDH) was used as a loading control to which protein expression was normalized. (mean ± SEM, one-way ANOVA, Tukey's post hoc, *n* = 3 biologically independent animals/group examined over three independent experiments).

0.05) exclusively following EE + SNA, compared to EE Sham or SH SNA alone (Fig. 2d). To validate the upregulation of NOX2 complex, we performed quantitative reverse-transcriptase PCR (RT-PCR) and confirmed that the subunits p22phox (*Cyba*), gp91phox (*Cybb*), p47phox (*Ncf1*), p67phox (*Ncf2*), and p40phox (*Ncf4*) were significantly upregulated following EE + SNA, with a similar trend for Rac1 (Supplementary Fig. 2). Next, we analyzed the protein expression of the NOX2 complex components using immunoblotting analysis from DRG, which confirmed a significant upregulation of p22phox (*CYBA*), gp91phox (*CYBB*), and p47phox (*NCF1*) following EE + SNA (Fig. 2e–f). The p40phox and p67phox subunits of the NOX2 complex could not be investigated due to the lack of suitable antibodies. In addition, we analyzed the activation of NOX2 complex by immunoblotting of the regulatory subunit phospho-p47phox (p-p47phox), a marker of NOX2 activation. We found that p-p47phox was exclusively increased after EE + SNA (Fig. 2e, f). Furthermore, in line with these data, immunostaining DRG neurons for the structural subunit gp91phox, the regulatory subunit p47phox, as well as its

active phosphorylated state (p-p47phox) showed a significant increase in fluorescence intensity following EE + SNA (Supplementary Fig. 3). These findings support and extend the RNAseq data, suggesting that the NOX2 complex is upregulated and activated in DRG neurons after enriched conditioning.

**Neuronal NOX2 is required for enriched conditioning-mediated regeneration**. Thus far, we have reported an upregulation and activation of NOX2 in DRG neurons at both the gene and protein level after EE + SNA. Next, we asked whether a functional NOX2 complex is required for the EE + SNA-dependent increase in neurite outgrowth and regeneration of sensory axons. To functionally disrupt the NOX2 complex, we conditionally deleted the structural subunit gp91phox (*CYBB*) in DRG neurons, by injecting AAV5-Cre-GFP or AAV5-GFP as a control in the sciatic nerves of gp91phox[fl/fl] mice (Supplementary Fig. 4). Four weeks post injection, the mice were exposed to SH or EE for 10 days before performing Sham or SNA. Twenty-four hours after Sham or SNA, sciatic DRG neurons were dissected

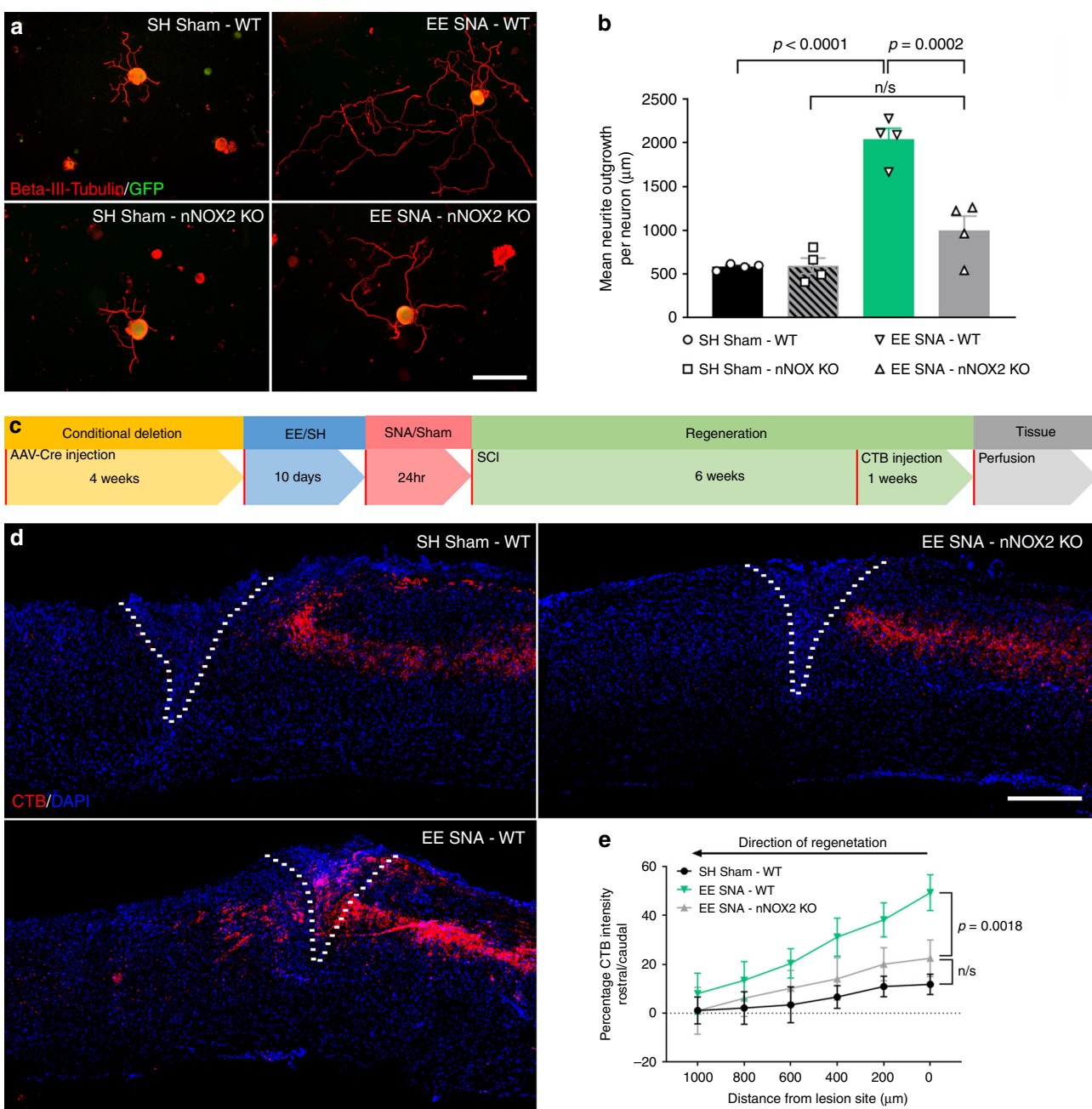

**Fig. 3 NOX2 is required for EE + SNA-dependent increase in neurite outgrowth and axon regeneration after SCI. a** Representative images of cultured DRG from NOX2$^{fl/fl}$ mice previously transduced in vivo with AAV-GFP or AAV-Cre-GFP (green) and stained with Beta-III-tubulin (red) after SH Sham or EE + SNA. Scale bar, 100 μm. **b** Quantification of average neurite outgrowth per neuron (mean ± SEM, one-way ANOVA, Tukey's post hoc, n.s. = nonsignificant, $n = 4$ biologically independent animals/group, average of 20 cells/replicate). **c** Timeline for the in vivo experiment. **d** Representative images of CTB-traced (red) dorsal column sensory axons after injury and DAPI (blue) to determine the lesion site (dashed line). Scale bar, 400 μm. **e** Quantification of CTB-positive regenerating axons (mean ± SEM, two-way repeated-measures ANOVA, Tukey's post hoc, n.s. = nonsignificant, $n = 5$ biologically independent animals/group). Fluorescence intensity was measured in one series of tissue for each spinal cord.

and cultured for 12 h. Analysis of neurite outgrowth from green fluorescent protein (GFP)-positive DRG showed that NOX2 deletion significantly attenuated the EE + SNA-dependent increase in outgrowth (Fig. 3a, b), suggesting that a functional NOX2 complex is required for the EE + SNA-dependent increase in regenerative potential. To determine whether functional NOX2 is required for EE + SNA-dependent sensory axon regeneration in vivo, mice with a conditional deletion of gp91phox were exposed to EE + SNA or SH Sham before receiving a dorsal spinal hemisection. Six weeks later, regeneration of sensory axons was

assessed using the retrograde axonal tracer CTB, which was injected bilaterally in the sciatic nerve 1 week before sacrificing the mice (Fig. 3c). Exposure to EE + SNA significantly increased the number of sensory axons into and past the spinal lesion site (Fig. 3d, e), without affecting the astrocytic scar around the lesion site (Supplementary Fig. 5). Disrupting the function of the NOX2 complex by deleting gp91phox abolished the enriched conditioning (EE + SNA)-dependent axonal regeneration to the level observed in SH Sham conditions, where the axonal front retracted from the lesion border (Fig. 3d, e).

Recently, we reported that conditional deletion of NOX2 in macrophages blocks the SNA-dependent increase in regenerative growth[15]. To investigate whether macrophage derived NOX2 is involved in the EE + SNA-dependent outgrowth, we cultured DRG neurons from mice where NOX2 was specifically deleted in macrophages (LysM-Cre/ NOX2[fl/fl] mice)[34]. We did not observe a significant reduction in neurite outgrowth in DRG neurons from macrophage NOX2$^{-/-}$ mice, suggesting that macrophage derived NOX2 is not involved in the enriched conditioning (EE + SNA) phenotype (Supplementary Fig. 6).

Taken together, these data show that neuronal NOX2 is required for the enriched conditioning (EE + SNA)-dependent increase in the regeneration of ascending sensory DRG axons.

**Neuronal NOX2 is required for enriched conditioning-dependent ROS production and redox signaling.** We next investigated whether exposure to EE + SNA resulted in a biologically active NOX2 complex-dependent increase in reactive oxygen species (ROS) in the DRG. We characterized the temporal regulation of ROS production following EE + SNA or SH Sham before SCI, as well as at 3, 7, and 42 days after SCI. DRG were extracted and incubated with hydrocyanine–Cy3, a cell permeable pan-ROS probe. The oxidation of the probe results in the emission of red fluorescence, thus allowing for visualization of ROS levels. We found that exposure to EE + SNA enhanced ROS compared to SH Sham (Fig. 4a, b) in line with an increase in NOX2 expression identified by RNAseq (Fig. 2d), RT-PCR (Supplementary Fig. 2), and immunoblotting (Fig. 2e, f). The increase in hydrocyanine–Cy3 after EE + SNA compared to SH Sham was also observed until 3 days after SCI (Supplementary Fig. 7). Importantly, the elevation in hydrocyanine–Cy3 signal was blocked in DRG from mice where NOX2 (gp91phox[fl/fl]) was conditionally deleted at all the time points analyzed (Fig. 4a, b and Supplementary Fig. 7).

To shed light on the downstream effects of the NOX2-dependent redox signaling, we performed redox proteomics using BIAM switch assay in combination with mass spectrometry in wild-type (WT) and NOX2$^{-/-}$ DRG after EE + SNA or SH Sham. Genetic deletion of NOX2 (NOX2$^{-/-}$) led to the reduced oxidation of 484 out of 505 differentially oxidized proteins after enriched conditioning (EE + SNA) and 476 out of 500 after SH Sham (Supplementary Data 6). In addition, we found that NOX2 is required for the oxidation of 205 proteins specifically after enriched conditioning (EE + SNA) and 197 proteins after SH Sham (Supplementary Data 6). Next, we performed GO functional analysis (ClueGO in Cytoscape) of the EE + SNA-specific or SH Sham-specific differentially oxidized proteins. The analysis showed that the most represented functional categories after EE + SNA were related to response to injury and wound-healing, signaling, lipid metabolism, extracellular matrix, and cell adhesion (Fig. 4c, Supplementary Fig. 8, and Supplementary Data 7). However, in SH Sham animals, the most represented categories were related to mitochondrial metabolism, cellular respiration, transmembrane transport, and proteasome (Fig. 4c, Supplementary Fig. 8, and Supplementary Data 7).

Together, these data show that in addition to a role in the axonal regeneration of DRG sensory fibers, NOX2 is required for ROS production and activation of specific redox signaling pathways following enriched conditioning.

**Active STAT3 is implicated in NOX2 complex expression and enriched conditioning-dependent axon regeneration via PKC signaling.** As we observed an EE + SNA-dependent increase in the expression of the NOX2 complex at the mRNA and protein level, we investigated which TFs might be responsible to drive the

expression of the NOX2 complex subunits. In silico TF analysis of the NOX2 subunit promoters[35] identified STAT3 as the highest ranked TF potentially implicated in NOX2 transcriptional regulation (Fig. 5a). We therefore assessed STAT3 activation by immunoblotting analysis of phosphorylated STAT3 (pSTAT3) in DRG after EE + SNA, which showed a significant increase compared to SH Sham (Fig. 5b, c). To ascertain whether pSTAT3 occupies the promoters of NOX2 complex in vivo, we performed pSTAT3 chromatin immunoprecipitation (ChIP) from DRG followed by quantitative PCR (qPCR) analysis of the promoter regions of NOX2 complex after exposure to EE + SNA or SH Sham. We found that EE + SNA significantly increased pSTAT3 occupancy on all of the five NOX2 complex subunits (*Cyba*, *Cybb*, *Ncf1*, *Ncf2*, and *Ncf4*) compared to SH Sham (Fig. 5d). Successful TF occupancy and transcriptional activation is associated with histone acetylation at gene promoters[36,37]. We have previously shown that exposure to EE strongly induces histone acetylation in DRG neurons including at H3K27, a mark of transcriptional activation[30]. Therefore, we hypothesized that combining EE + SNA induces histone acetylation at NOX2 promoter regions facilitating STAT3 occupancy, thereby increasing the transcription of NOX2 complex genes. Indeed, pre-exposure to EE + SNA enhanced the acetylation of H3K27 in DRG neurons (Supplementary Fig. 9). ChIP for H3K27ac showed significant enrichment at three of the same promoters (*Cyba*, *Ncf1*, and *Ncf2*) occupied by pSTAT3 with a similar trend for *Cybb* ($p = 0.08$) (Fig. 5e). These data show that pSTAT3 occupies hyperacetylated NOX2 promoter regions after enriched conditioning (EE + SNA).

To characterize whether active STAT3 and histone acetylation are required to drive gene expression of NOX2 complex genes, HEK293T cells were transfected with a constitutively active (CA), dominant negative, or WT-STAT3, in the presence or absence of the pan-histone deacetylase inhibitor panobinostat, which increases histone acetylation. We found that only the CA STAT3 in combination with panobinostat was able to induce the expression of all NOX2 complex genes, suggesting that STAT3 activation is necessary to induce transcription of hyperacetylated NOX2 genes (Supplementary Fig. 10).

To establish whether neuronal STAT3 is involved in the enriched conditioning (EE + SNA)-dependent increase in axon regeneration in vivo, we conditionally deleted STAT3 in DRG neurons by injecting an AAV-Cre-GFP or control AAV-GFP in the sciatic nerves of STAT3[fl/fl] mice 4 weeks before performing a dorsal spinal hemisection. To assess sensory axon regeneration, CTB tracer was injected bilaterally in the sciatic nerves 1 week before killing the mice (Fig. 6a). Deletion of STAT3 in DRG neurons (Supplementary Fig. 11) significantly reduced the enriched conditioning (EE + SNA) effect on axon regeneration after SCI (Fig. 6b, c) without affecting the astrocytic scar (Supplementary Fig. 12), similar to what was observed after conditional deletion of NOX2 (gp91phox). Moreover, we found that STAT3 conditional deletion significantly reduced the EE + SNA-mediated expression of gp91phox in AAV-Cre-GFP-positive DRG compared to AAV-GFP-injected controls (Supplementary Fig. 13). Altogether, although we do not exclude the additional contribution of other TFs, these data show that following enriched conditioning (EE + SNA), active STAT3 is a key TF responsible for NOX2 complex expression and plays an important role in the EE + SNA-mediated increase in sensory axon regeneration.

We next investigated which signaling pathways might contribute to the phosphorylation and activation of STAT3 following EE + SNA. The RNAseq GO analysis showed that DAG/PKC and GTPase signaling were the highest enriched pathways after EE + SNA (yellow box Fig. 2b) and the protein–protein interaction network showed they are functionally

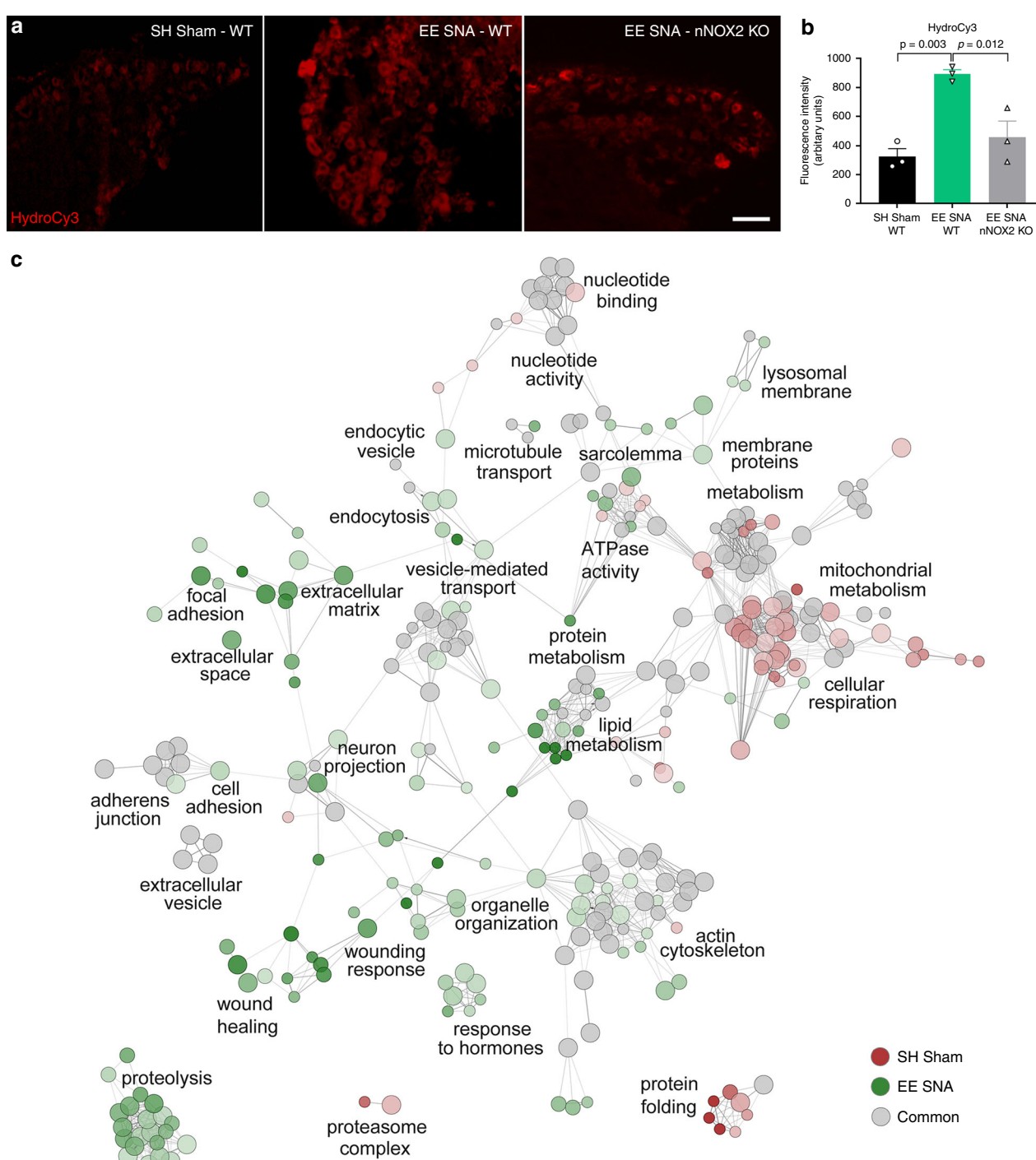

**Fig. 4 NOX2 is required for EE + SNA-dependent increase in ROS production and redox signaling in DRG. a** Representative images of DRGs from WT or NOX2[fl/fl] mice after EE SNA or SH Sham, hydrocyanine–Cy3 (red) staining of ROS 1 day after SCI. Scale bar, 100 μm. **b** Quantification of hydrocyanine–Cy3 in DRG neurons (mean ± SEM, one-way ANOVA, Tukey's post hoc, $n = 3$ biologically independent animals/group, examined over three independent experiments). Fluorescence intensity was measured in one series of tissue for each DRG. **c** Network visualization of the GO clustering analysis of the differentially oxidized proteins in EE + SNA (green) and SH Sham (red)-specific datasets run with ClueGO in Cytoscape (Bonferroni *P*-value < 0.05). Each node represents a GO-enriched term and the size of each node represents its enrichment significance. Edges represent interrelations between terms, defined by the Kscore. Red and green color intensity reflects proteins from a given dataset (red = Sham, green = EE + SNA, gray = common). Main functional groups are reported.

connected to NOX signaling (Fig. 2c). Given that DAG and GTPases are classical activators of PKC[38], which can phosphorylate STAT3[39,40], we hypothesized the involvement of PKC signaling in EE + SNA-dependent STAT3 phosphorylation. Indeed, immunofluorescence experiments showed that pPKC

expression in DRG neurons was significantly higher following EE + SNA compared to SH Sham (Supplementary Fig. 14). Next, we assessed whether PKC activity is required for the EE + SNA-mediated phosphorylation of STAT3 and the increase in DRG neurite outgrowth. We modulated PKC activity by delivering the

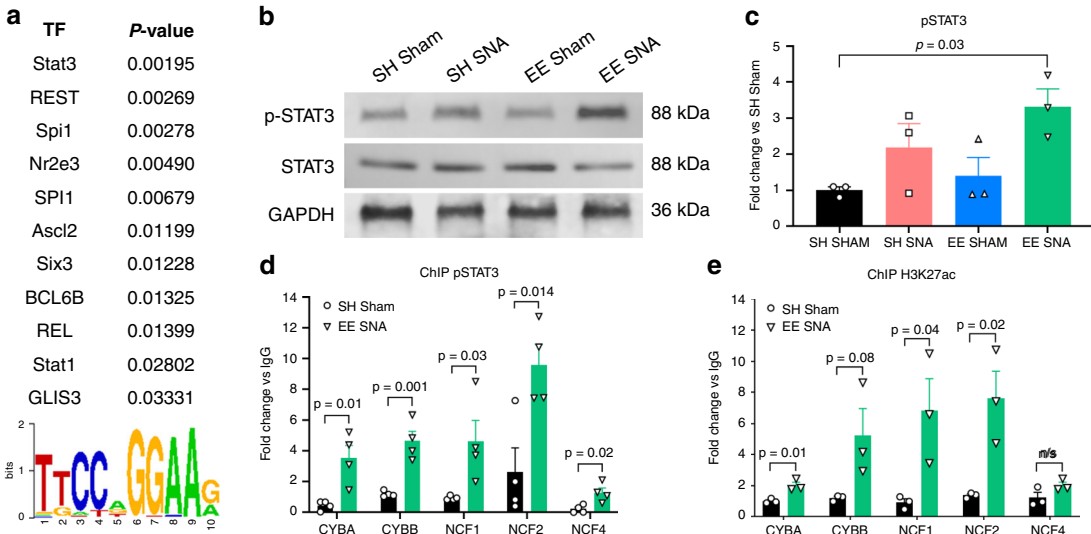

**Fig. 5 pSTAT3 is induced and occupies the promoters of NOX2 subunits following EE + SNA. a** In silico motif analysis run with Pscan of the NOX2 gene member's promoters, showing the putative transcription factors regulating their transcription ranked by *p*-value. STAT3 binding motif; letters represent the relative enrichment for each gene at that position. **b** Immunoblotting of pSTAT3 from DRG extracts after SH Sham, EE Sham, SH SNA, or EE + SNA. **c** Quantification of immunoblotting, pSTAT3 was normalized to levels of STAT3, GAPDH was used as a loading control (mean ± SEM, one-way ANOVA, Tukey's post hoc, $n = 3$ biologically independent animals/group examined over three independent experiments). **d** Quantitative RT-PCR analysis of NOX2 complex genes after chromatin immunoprecipitation (ChIP) for pSTAT3, expressed as fold change of DNA enrichment compared to IgG (mean ± SEM, two-tailed unpaired Student's *T*-test, $n = 4$ independent biological replicates/group). **e** Quantitative RT-PCR analysis of NOX2 complex genes after ChIP for H3K27ac, expressed as fold change of DNA enrichment compared to IgG (mean ± SEM, two-tailed unpaired Student's *T*-test, n.s. = nonsignificant, $n = 3$ independent biological replicates/group, examined over three independent experiments).

small molecule activator Ingenol 3-angelate (I3A) or inhibitor Gö 6983 to DRG neurons cultured from mice exposed to SH Sham or EE + SNA. Activation of PKC using I3A significantly enhanced DRG neurite outgrowth and phosphorylation of STAT3 (Fig. 6d–f). Conversely, PKC inhibition with Gö 6983 significantly reduced the EE + SNA-dependent increase in neurite outgrowth and STAT3 phosphorylation (Fig. 6d–f). Taken together, these data show that enriched conditioning (EE + SNA) requires PKC activity to phosphorylate STAT3 and increase neurite outgrowth of DRG neurons.

**AAV-mediated overexpression of CA p47phox in DRG neurons induces axon regeneration and synaptic plasticity after SCI.** Lastly, we investigated whether activating the NOX2 complex is able to promote sensory axon regeneration and functional recovery following SCI. To activate neuronal NOX2 signaling, we expressed a CA phospho-mimetic form of p47phox (p47-3xS\D mutant GFP)[15,41] or AAV-GFP in DRG neurons by AAV delivery into the sciatic nerves (Supplementary Fig. 15). To rule out the possibility that virus injection affects the nerve micro-environment, we analyzed sciatic nerve injection sites and found that very little GFP expression co-localized with the monocyte/ macrophage marker CD68 or Schwann cell marker SOX10 (Supplementary Fig. 16A–D). However, we observed ~80% colocalization of GFP with the axonal marker TUJ1 (Supplementary Fig. 16E, F), suggesting that AAV8 preferentially transduces neurons and not Schwann cells or monocytes/macrophages after injection in the sciatic nerve.

A T9 dorsal spinal hemisection was performed 4 weeks after AAV transduction and the mice were killed 6 weeks after the SCI (Fig. 7a). Anatomical analysis showed multiple p47phox-GFP-positive axons across and rostral to the lesion site as opposed to the GFP control animals whose axons were not found in the lesion site (Fig. 7b–d). No difference in GFAP intensity, a marker of the astrocytic scar that surrounds the lesion site, was detected

between the groups (Supplementary Figure 17). In addition, we examined coronal sections at the T2 spinal level to visualize spared fibers above the lesion: any animals with spared fibers were removed from the analysis (Supplementary Fig. 18).

In addition to promoting axon growth, p47-3X-GFP over-expression also increased the number of vesicular gamma-aminobutyric acid transporter (vGAT)- and vesicular glutamate transporter 1 (vGlut1)-positive boutons (putative inhibitory and excitatory synapses) apposed to choline acetyl transferase-positive motor neurons in the lumbar spinal cord below the injury, as shown by multi-fluorescent orthogonal three-dimensional (3D) confocal images (Fig. 7e–h). Taken together, this data suggests spinal circuit reorganization and synaptic plasticity between motor neurons and both inhibitory propriospinal neurons (vGAT) and excitatory neurons, specifically group-Ia proprio-ceptive afferents (vGlut1), which have previously been associated with improvements in functional recovery[42–44]. To address whether overexpression of p47-3X was able to promote recovery of sensory function, we carried out the adhesive tape test that measures the time the animals take to first contact and then remove an adhesive tape placed on their hind paws. Indeed, we found that, by 6 weeks after the injury, p47-3X overexpression induced a significant improvement in both the time to sense and to remove the tape (Fig. 7i, j). Taken together, these data suggest that AAV-mediated overexpression of active NOX2 promotes plasticity and growth of sensory axons, as well as improvements in sensory function after SCI.

## Discussion
In this study, we describe an experimental model that we call "enriched conditioning" that operates via a neuronal intrinsic regenerative redox mechanism. Enriched conditioning significantly potentiates the regenerative ability of DRG neurons compared to the classical conditioning injury paradigm that has represented the gold standard for sensory axon regeneration since

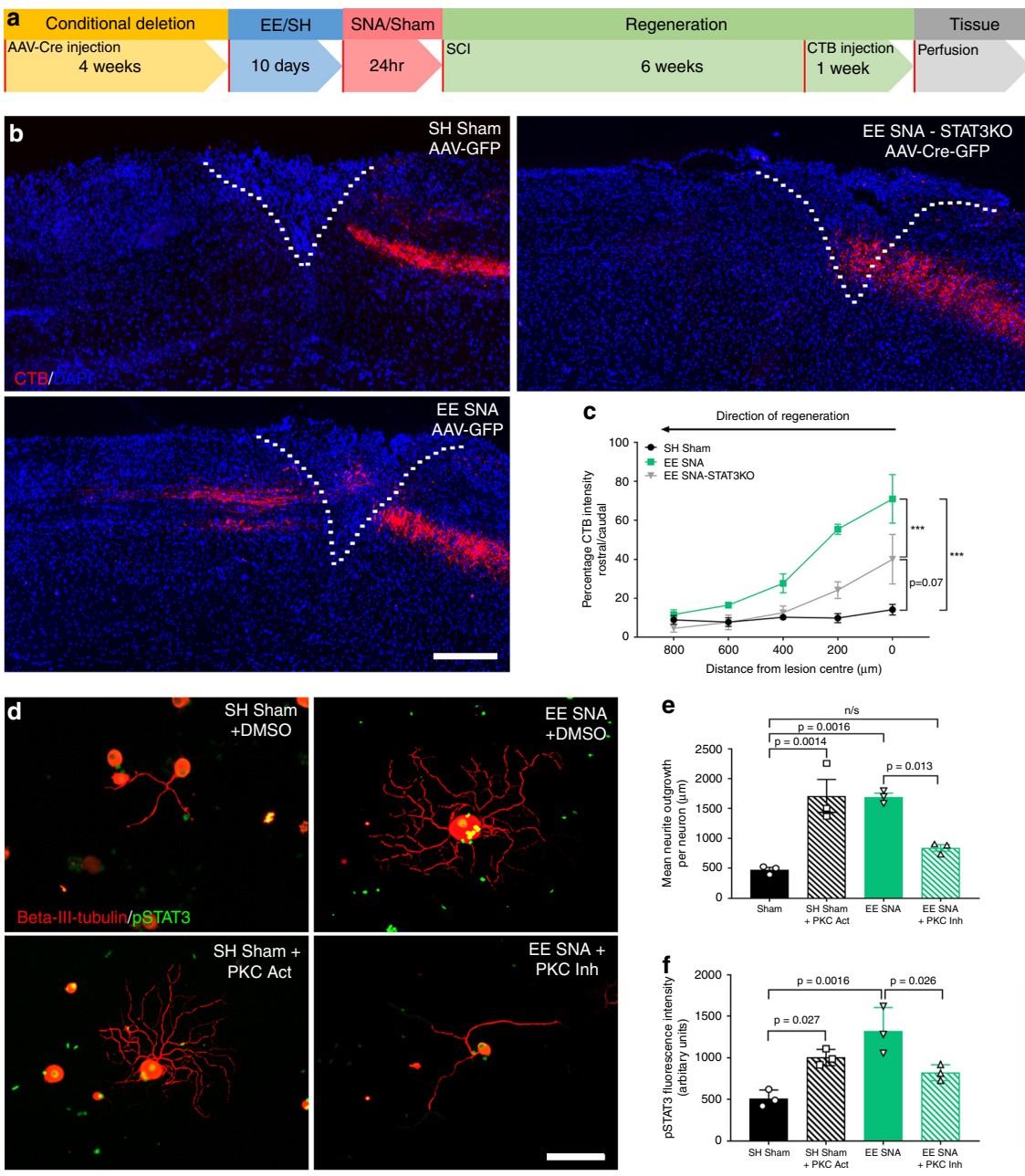

**Fig. 6 STAT3 deletion in DRG blocks EE + SNA-dependent axonal regeneration and NOX2 complex expression. a** Timeline for the in vivo experiment. **b** Representative images of WT (injected with control AAV-GFP) and STAT3 KO (injected with AAV-Cre-GFP) CTB-traced (red) dorsal column sensory axons after injury and DAPI (blue), to determine the lesion site (dashed line). Scale bar, 200 μm. **c** Quantification of fluorescence intensity of CTB-positive regenerating axons (mean ± SEM, two-way repeated-measures ANOVA, Tukey's post hoc, \*\*\*P-value < 0.0001, n = 4 biologically independent animals/group). Fluorescence intensity was measured in one series of tissue for each spinal cord. **d** Representative images of cultured DRG neurons from SH Sham + vehicle (DMSO), SH Sham + PKC activator, EE + SNA + vehicle (DMSO), and EE + SNA + PKC inhibitor, stained with Beta-III-tubulin (red) and pSTAT3 (green). Scale bar, 100 μm. **e** Quantification of mean neurite outgrowth per neuron (mean ± SEM, one-way ANOVA, Tukey's post hoc, n.s. = nonsignificant, n = 3 biologically independent animals/group, average of 20 cells/replicate). **f** Quantification of pSTAT3 fluorescence intensity (mean ± SEM, one-way ANOVA, Tukey's post hoc, n = 3 biologically independent animals/group, average of 20 cells/replicate).

its discovery several decades ago[18–20]. We found that combining injury-independent (EE) and injury-dependent (SNA) approaches induced an additive effect on the regenerative potential of DRG neurons, displaying redox-mediated enhanced regeneration of ascending sensory axons well beyond the lesion site. The degree of regeneration elicited by enriched conditioning represents a superior standard for the regenerative ability of sensory neurons following a SCI for which we have begun to characterize the underlining molecular mechanisms. Specifically, we found

that enriched conditioning induces PKC-dependent phosphorylation of STAT3, which drives the expression of NOX2 complex subunits by occupying hyperacetylated NOX2 promoter regions. We next demonstrated that activated STAT3 is important for NOX2 expression in DRG neurons and for enhanced regenerative potential elicited by enriched conditioning. We also showed that expression of a functional NOX2 complex in DRG neurons is required for the enriched conditioning-dependent increase in axon regeneration, as well as for redox signaling. Lastly, and

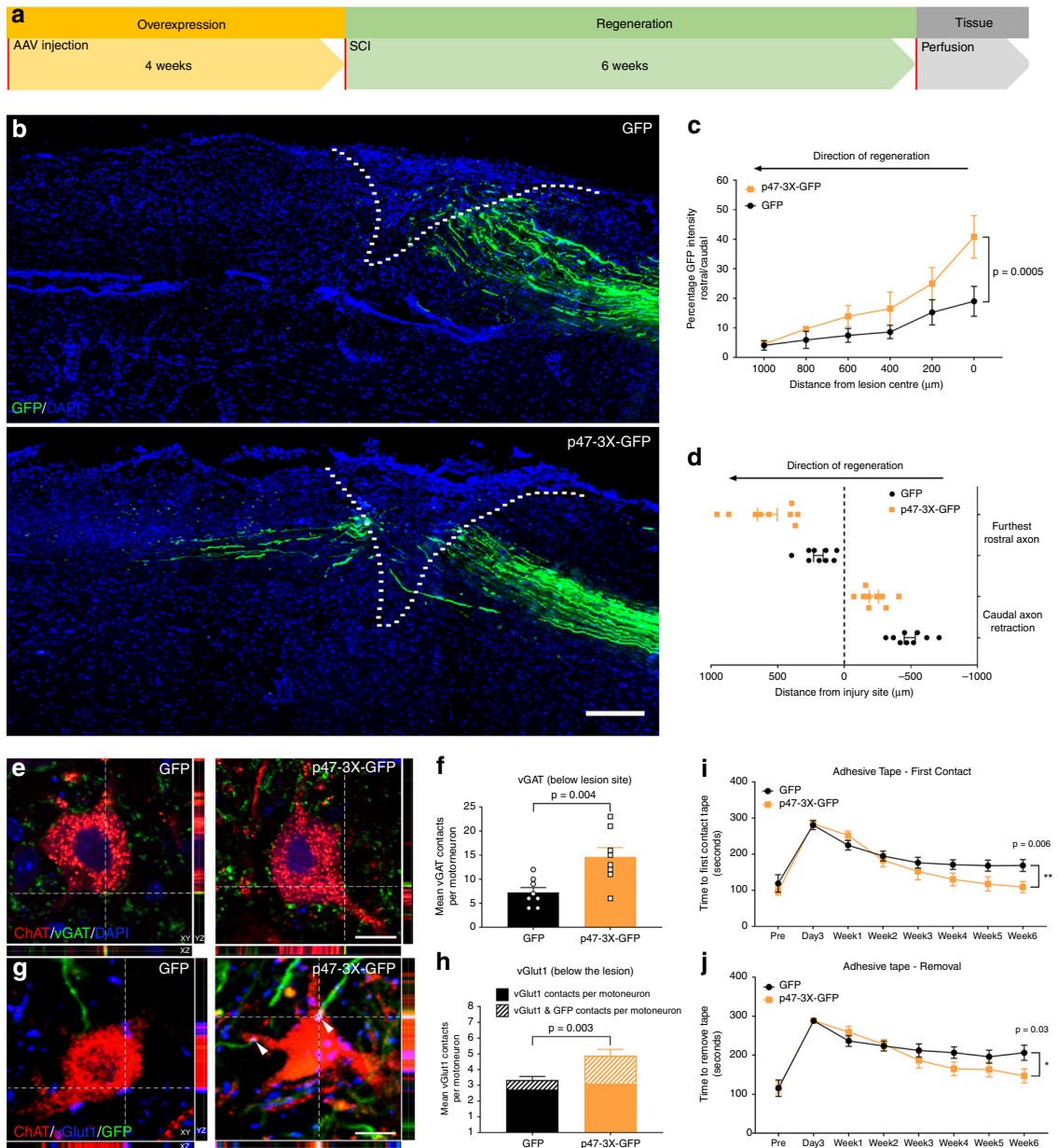

**Fig. 7 Overexpression of constitutively active p47phox (p47-3X) in DRG neurons induces axon regeneration and synaptic plasticity after SCI.**
**a** Timeline for the in vivo experiment. **b** Representative images of GFP-positive sensory axons (green) after spinal cord injury and DAPI (blue) to determine the lesion site (dashed line). Scale bar, 200 μm. **c** Quantification of fluorescence intensity of GFP-positive regenerating axons (mean ± SEM, two-way repeated-measures ANOVA, Tukey's post hoc, $n = 9$ biologically independent animals/group examined over two independent experiments). Fluorescence intensity was measured in one series of tissue for each spinal cord. **d** Average distance between lesion center and furthest regenerating axon/caudal tract retraction (mean ± SEM, $n = 9$ biologically independent animals/group examined over two independent experiments. Length of the furthest regenerating five axons and main tract retraction was measured in one series of tissue for each spinal cord). **e** Multi-fluorescent orthogonal 3D confocal images show colocalization between vGAT-positive boutons (green) and ChAT-positive motor neurons (red) below the injury (L1-4). Scale bar 25 μm, the orthogonal planes XY, XZ, and YZ are labeled. **f** Quantification of vGAT-positive boutons apposed to motoneurons (mean ± SEM, two-tailed unpaired Student's $t$-test, $n = 9$ biologically independent animals/group, average of 20 motoneurons/replicate examined over two independent experiments). **g** Multi-fluorescent orthogonal 3D confocal images show juxtaposition between vGlut1-positive boutons (blue), GFP-positive axons (green), and ChAT-positive motoneurons (red) below the injury (L1-4), arrowheads indicate triple colocalization. Scale bar 25 μm, the orthogonal planes XY, XZ, and YZ are labeled. **h** Quantification of vGlut1-positive boutons apposed to motoneurons (whole bar), vGlut1 and GFP colocalization apposed to motoneurons (cross-hatched area) (mean ± SEM, two-tailed unpaired Student's $t$-test, $n = 9$ biologically independent animals/group, average of 20 motoneurons/replicate examined over 2 independent experiments). **i** Quantification of the time required to first contact an adhesive pad placed on the hind paws (mean ± SEM, two-way repeated-measures ANOVA, Tukey's post hoc, $n = 9$ biologically independent animals/group examined over two independent experiments). **j** Quantification of the time required to remove an adhesive pad placed on the hind paws (mean ± SEM, two-way repeated-measures ANOVA, Tukey's post hoc, $n = 9$ biologically independent animals/group examined over two independent experiments).

importantly, AAV-mediated overexpression of active NOX2 is able to promote plasticity and regeneration of sensory axons after SCI along with partial recovery of sensory function.

The role of the PKC signaling pathway in axon regeneration is still ambiguous, with multiple studies reporting contradictory findings. In line with our results, several previous studies showed that increased PKC signaling is associated with sciatic nerve regeneration and that overexpression of specific PKC isoforms can promote DRG outgrowth as well as axon regeneration of retinal ganglion cells[16,45,46]. These findings support the evidence that enriched conditioning enhances PKC activation and a downstream increase in STAT3 phosphorylation, which are important for increasing the regeneration potential of DRG neurons. However, in apparent contradiction to our results, it has also been shown that intrathecal infusion of another PKC inhibitor (Gö 6976) induces regeneration of descending corticospinal tract axons as well as of ascending sensory fibers[47,48]. As the cellular concentration of these inhibitors can be hard to control in vivo, it is difficult to interpret these results with regard to the specificity of PKC isoform inhibition or of other kinases[49–52]. In addition, the PKC signaling pathway may play different regenerative roles in specific neuronal populations and the time at which the pathway is activated or inhibited after injury might affect its role in signaling and regeneration.

STAT3 phosphorylation is critical for its activation and for its ability to support axonal regeneration[53–55]. However, this is the first evidence of STAT3-dependent NOX2 expression promoting the regeneration of sensory axons, highlighting the importance of neuronal derived NOX2 signaling. In turn, the evidence that STAT3 is important for NOX2 expression further emphasizes the central role that this TF plays in axon regeneration. STAT3-dependent NOX2 expression requires the acetylation of NOX2 promoters, indicating that permissive chromatin states might facilitate pSTAT3-dependent induction of additional regeneration associated genes.

Furthermore, our data provide a number of additional TFs and signaling pathways that might be involved in enriched conditioning requiring further confirmatory studies that will expand the array of regenerative targets to promote repair and recovery after SCI.

Although a conditioning lesion promotes macrophage-dependent exosomal release of NOX2 complexes into injured axons[15], enriched conditioning triggers intrinsic neuronal expression of NOX2 complex that leads to significantly increased axonal regeneration after SCI compared to conditioning alone. The superior regenerative capacity following enriched conditioning might be explained by the intrinsic neuronal transcriptional activation of NOX2 that is further supported by epigenetic mechanisms such as histone acetylation at NOX2 promoters. These epigenetic mechanisms likely allow increased transcriptional activity over prolonged periods of time, as suggested by recent studies where the role of epigenetic regulation in axonal regeneration has been reported[30,56–63]. In line with this, we found that enriched conditioning induces an enduring increase in NOX2-dependent redox signaling as suggested by elevated ROS levels up to one week following SCI.

To elucidate NOX2-dependent redox signaling induced by enriched conditioning, we chose an unbiased systematic approach by using state of the art redox proteomics. Functional analysis of the differentially redox modified proteins indicated that NOX2 was required for pathways involved in cell signaling and wound-healing response exclusively after enriched conditioning, where NOX2 is needed for axonal regeneration. It also suggests a NOX2-dependent shift from mitochondrial metabolism and respiration to redox signaling, directed to cellular repair and wound-healing processes.

This study provides evidence for the existence of neuronal intrinsic regenerative redox signaling and it demonstrates that AAV-mediated activation of neuronal NOX2 promotes axon regeneration, synaptic plasticity, and partial recovery of sensory function after SCI. While this work suggests potentially druggable targets within the PKC-STAT3-NOX2 signaling axis, it also advocates against the indiscriminate use of post-injury anti-oxidant treatment.

In summary, we have discovered a model whereby sensory neurons convey injury-dependent and injury-independent signals onto intracellular molecular pathways that promote regenerative redox signaling in the CNS, overcoming the current standards set by the conditioning lesion. Similar to the classical conditioning paradigm that, over the years, allowed the identification and targeting of a number of regenerative pathways, we believe that the enriched conditioning model will represent an improved platform for the discovery of regenerative molecular mechanisms and targets.

## Methods

**Mice.** Animal work was carried out in accordance to regulations of the UK Home Office under the Animals (Scientific Procedures) Act 1986, with Local Ethical Review by the Imperial College London Animal Welfare and Ethical Review Body Standing Committee (AWERB). B6(Cg)-$Ncf1^{m1J}$/J (Ncf1$^{-/-}$, Jackson), B6.129S-$Cybb^{tm1Din}$/J (gp91phox$^{-/-}$/NOX2$^{-/-}$, Jackson), B6(Cg)-Cybbtm1.1Abk/J (NOX2$^{fl/fl}$,[34]), B6.129S1-Stat3tm1Xyfu/J (STAT3$^{fl/fl}$, Jackson), LysM-Cre-NOX2$^{-/-}$, and corresponding WT littermates or C57Bl6/J (Harlan, UK) mice ranging from 4 to 8 weeks of age were used for all experiments. For generation of macrophage-specific NOX2$^{-/-}$ mice, NOX2$^{fl/fl}$ mice were crossed with LysM-Cre mice as previously described[15]. For all surgeries, mice were anesthetized with isoflurane (5% induction 2% maintenance) and a mixture of buprenorphine (0.1 mg/kg) and carprophen (5 mg/kg) was administered peri-operatively as analgesic.

**Animal housing.** Animals were kept on a 12 h light : dark cycle with food and water provided ad libitum, at a constant room temperature (RT) and humidity (21 °C and 5%, respectively). Standard housing for mice consisted of 26 × 12 × 18 cm$^3$ cages housing four mice with tissue paper for bedding, a tunnel and a wooden chew stick. The enriched environment housing consisted of 36 × 18 × 25 cm$^3$ cages housing eight mice with tissue paper for bedding, a tunnel and a wooden chew stick. EE cages also received additional nesting material which included nestlets, rodent roll and sizzle pet (LBS Biotech). EE cages continually contained a hanging plastic tunnel (LBS Biotech) and a plastic igloo combined with a fast-track running wheel (LBS Biotech). In addition to this, EE cages received a wooden object (cube, labyrinth, tunnel, corner 15) (LBS Biotech), which was changed after 5 days, to help maintain a novel environment. The EE cages also received 15 g fruity gems (LBS Biotech) every 5 days, to encourage exploratory and natural foraging behavior.

**Spinal cord injury.** Surgeries were performed as previously reported. A laminectomy at vertebra T9 was performed to expose spinal level T12 and a dorsal hemisection until the central canal was then performed using microscissors (Fine Science Tools). For the sham surgery, a laminectomy was performed but the dorsal hemisection was not. Any bleeding was stopped and the muscle and skin was sutured closed. Six weeks after SCI, animals were deeply anesthetized and perfused transcardially.

**Sciatic nerve axotomy.** Briefly, sciatic nerve lesion experiments were performed under isoflurane anesthesia (5% induction, 2% maintenance). The biceps femoris and the gluteus superficialis were separated by blunt dissection, and the sciatic nerve was exposed. Sciatic injury was performed by complete transection using microscissors. Sham-operated mice that underwent exposure of the sciatic nerve without axotomy were used as surgery controls.

**Local delivery of compounds or viral particles on the sciatic nerve.** For local compound delivery, 2.5 μl of each compound or its corresponding vehicle (saline or dimethyl sulfoxide (DMSO)) were slowly poured onto the nerve with a pipette to avoid damage on the nerve and allowed to penetrate the tissue for several minutes prior performing the injury or the sham procedure before suturing the wound. For retrograde tracing and transport and experiments, 3 μl of Dextran-584 (Life Technology) or 1% CTB (List Biological Laboratories) were injected into the nerve prior to dorsal column axotomy or sciatic nerve crush. For AAV virus delivery to DRG neurons, 2.5 μl of AAV5-GFP (SignaGen Laboratories), or AAV5-Cre-GFP (SignaGen Laboratories), or AAV8-GFP (Viral Vector Core, The Miami Project) or AAV8-p47-3X-GFP (Viral Vector Core, The Miami Project) were injected into the

sciatic nerve with a 10 µl Hamilton syringe and Hamilton needle (NDL small RN ga34/15 mm/pst45°) (Hamilton) before suturing the wound.

**Pharmacological compounds**. PKC inhibitor Gö 6983 (ab144414) and PKC activator I3A (ab144280) were purchased from Abcam and Panobinostat was purchased from Selleckchem. Solutions of 2 µM Gö 6983, 500 nM I3A, and 200 µM Panobinostat were dissolved in DMSO. For each group treated with a drug, the respective control group received the same volume of vehicle (DMSO).

**Western blotting**. Proteins from sciatic DRG were extracted using RIPA buffer with protease and phosphatase inhibitor cocktails (Roche). Total lysates were obtained by 30 min lysis on ice followed by 30 min centrifugation at 4 °C. Protein concentration of lysate was quantified using Pierce BCA Protein Assay Kit (Thermo Scientific). Ten to 50 µg proteins were loaded to SDS-polyacrylamide gel electrophoresis gels and transferred to polyvinylidene difluoride membranes for 2 h. Membranes were blocked with 5% bovine serum albumin (BSA) or milk for 1 h at RT and incubated with gp91phox (1:200, Mouse, BD Biosciences), p47phox (1:200, Rabbit, Santa Cruz), p-p47phox (S345) (1:100, Rabbit, Sigma), p67phox (1:500, Rabbit, Abcam), p22phox (1:100, Rabbit, Abcam), H3 (1:1000, Rabbit, Abcam), STAT3 (1:500, Mouse, Cell Signaling Technology), pSTAT3 (1:1000, Rabbit, Cell Signaling Technology), pPKC (1:1000, Rabbit, Cell Signaling Technology), PKC (1:1000, Rabbit, Abcam), or GAPDH (1:1000, Rabbit, Cell Signaling Technology) at 4 °C O/N. Following incubation with horseradish peroxidase-linked secondary antibody (1:1000 anti-Rabbit or anti-Mouse GE Healthcare) for 1 h at RT, membranes were developed with ECL substrate (Thermo Scientific).

**DRG cell culture**. Adult DRG were dissected and collected in Hank's balanced salt solution on ice. DRG were transferred to a digestion solution 5 mg/ml Dispase II (Sigma), 2.5 mg/ml Collagenase Type II (Worthington) in Dulbecco's modified Eagle's medium (DMEM; Invitrogen) and incubated at 37 °C for 45 min with occasional mixing. Subsequently, DRG were transferred to media containing 10% heat inactivated fetal bovine serum (Invitrogen), 1× B27 (Invitrogen) in DMEM: F12 (Invitrogen) mix, and were manually dissociated by pipetting until no remaining clumps of DRG were observed. Dissociated cells were spun down, resuspended in media containing 1× B27 and Penicillin/Streptomycin in DMEM:F12 mix, and plated at 3500 cells/coverslip. The culture was maintained in a humidified atmosphere at 5% $CO_2$ in air at 37 °C. For ex vivo culture, mice underwent surgeries as previously described, and 24 h after surgery mice were killed and sciatic DRG were collected and cultured for 12 h, fixed, and stained.

**Plasmids**. pCDNA6.2emGFP was from Invitrogen. pEGFPc1humanp47phox was a gift of Professor Saito at the Laboratory of Molecular Pharmacology, Biosignal Research Center, Kobe University, Kobe, Japan[64]. All STAT3 plasmids generated contain the same pEGFP-C1 backbone, which was derived by excising *Sp1* from the pEGFP-SP1 vector plasmid, gift from Beatrice Yue (Addgene plasmid #39325). The pEGFP-SP1 vector was XhoI/KpnI digested in a sequential restriction digest releasing the 2355 bp *Sp1* gene to yield the pEGFP-C1 expression vector. Klenow fragment of DNA Polymerase I (New England Biolabs) was used to blunt ends. Expression vector for ligation with SD *STAT3* was blunted after XhoI digestion. Alkaline phosphatase (New England Biolabs) was used to dephosphorylate ends of vector. Mouse S727D mutant *STAT3* was amplified from pCDNA3-S727D-STAT3, gift from Jie Chen (Addgene plasmid #73364) using primers stat3F and stat3R. The 2393 bp amplicon was purified using the QIAquick PCR Purification Kit (Qiagen). The purified PCR product was first HindIII digested followed by Klenow fragment DNA Pol I blunting and KpnI digestion. The SD *STAT3* insert was ligated into the expression vector, pEGFP-C1, using T4 DNA Ligase and T4 DNA Ligase Buffer (10×) (New England Biolabs). pLEGFP-WT-STAT3 and pLEGFP-Y705F-STAT3 plasmids were gifts from George Stark (Addgene plasmids #71450 #71445). pLEGFP-WT-STAT3 and pLEGFP-Y705F-STAT3 plasmids were both digested with HindIII to release the 2337 bp mouse *STAT3* fragments. The WT-*STAT3* and YF-*STAT3* fragments were ligated into separate pEGFP-C1 expression vectors. Agarose gel electrophoresis was routinely used, after restriction digests and PCR amplification of SD *STAT3*. Agarose gel electrophoresis was carried out on a 1% agarose gel in TAE buffer. Ethidium bromide was added to agarose to a final concentration of 0.5 µg/mL, followed by separation at 100 V for 45–75 min. Gels were visualized using an Ultraviolet Transilluminator (UVP). Each *STAT3* plasmid construct was introduced into 100 µl of DH5α competent cells by heat shock transformation. The transformed cells were incubated in Lysogeny Broth (LB) medium supplemented with 1 M $MgCl_2$, 1 M $MgSO_4$, and 2 M glucose at 37 °C for 1 h with shaking. Cells were centrifuged at 3000 × *g* for 5 min and plated on LB agar plates with 100 µg/mL of kanamycin. Colonies were picked and cultured in LB medium and kanamycin (100 µg/mL) overnight at 37 °C with shaking. Plasmid DNA purification was performed following the QIAprep Spin Miniprep (Qiagen) protocol. After confirming the validity of *STAT3* plasmid constructs, the plasmid DNA was purified following the EndoFree Plasmid Maxi (Qiagen) protocol for a larger yield of DNA. pCDNA6.2emGFP was from Invitrogen and pEGFPc1humanp47phox[64] was a gift of Professor Saito at the Laboratory of Molecular Pharmacology, Biosignal Research Center, Kobe University, Kobe, Japan. The

S303D/S304D/S328D mutations (described in ref. [65] were inserted by site-directed mutagenesis (NEB Q5 site-directed mutagenesis kit following the manufacturer's instructions).

**In vivo redox-state detection in tissue sections**. In vivo redox levels were detected using hydrocyanine–Cy3 (ROSstar550, LICOR). Freshly dissected DRG were washed in ice-cold phosphate-buffered saline (PBS)–diethylenetriamine pentaacetic acid (DTPA), incubated with 100 µM hydrocyanine–Cy3 in PBS for 30 min at 37 °C, and then washed once with PBS–DTPA before being fixed in 4% paraformaldehyde (PFA).

**Oxiproteomics**. Six WT and six NOX2$^{-/-}$ mice were treated with EE+SNA or Sham. After 24 h, DRGs were dissected and immediately placed into 20% (w/v) trichloric acid, to preserve the thiol redox state. Redox proteomics was performed using the BIAM switch assay[66]. Detailed description of sample preparation, mass spectrometry and data analysis were deposited to the ProteomeXchange Consortium via the PRIDE partner repository[67] with the dataset identifier PXD018085. Quantitative mass spectrometry data (LFQ) represent oxidized proteins. Data were further analyzed by Perseus (v. 1.6.1.3). Contaminants and reversed identification were removed. Individual proteins per each protein group were included in the analysis. Missing values were replaced by background values from normal distribution. Student's *t*-test (*p*-value) and Bonferroni-based false discovery rate (multiple testing, *q*-value) was used to identify significant enriched proteins between experimental groups.

**ClueGO clustering analysis**. GO clustering analysis of the differentially oxidized proteins in EE SNA and Sham-specific datasets was performed with ClueGO in Cytoscape 3.7.1 using the following standard parameters: Bonferroni *P*-value < 0.05; GO tree interval min2, max8; GO selection min3, 2%; Kscore 0.4; mouse background.

**Histology and immunohistochemistry**. Tissue was post-fixed in 4% PFA (Sigma) and transferred to 30% sucrose (Sigma) overnight to cryoprotect, the tissue was then embedded in OCT compound (Tissue-Tek) and frozen at −80 °C. DRGs were sectioned at 10 µm thickness and mouse spinal cord at 20 µm using a cryostat (Leica). Immunohistochemistry on tissue sections was performed according to standard procedures. For all antibodies used, antigen retrieval was performed submerging the tissue sections in 0.1 M citrate buffer (pH 6.2) at 98 °C for 5 min. Next, tissue sections were washed with PBS to remove the excess of citrate buffer and blocked for 1 h with 8% BSA, 1% PBS-TX100. Finally, the sections were incubated with anti-p47phox (1:200, rabbit, Santa Cruz), p-p47phox (S345, 1:200, rabbit, Sigma), gp91phox (1:200, Mouse, BD Biosciences), pSTAT3 (1:200, Rabbit, Cell Signaling Technology 9145), CTB (1:1000, List biological 703), Tuj1 (1:1000, Promega G7121), H3K27ac (1:500, Abcam ab4729), GFAP (1:500, Millipore AB5804), CD68 (1:500, Abcam ab213363), vGlut1 (1:1000, Synaptic system 135302), vGAT1 (1:500, Synaptic Systems 131011), p-PKC (1:500, Cell Signaling Technology 9371), NF200 (1:1000 Millipore MAB5262), SOX10 (1:1000, Abcam ab264405) antibodies at 4 °C O/N. This was followed by incubation with Alexa Fluor-conjugated goat secondary antibodies according to standard protocol (1:1000, Invitrogen). Slides were counterstained with 4′,6-diamidino-2-phenylindole (DAPI) to visualize nuclei (1:5000, Molecular Probes).

**Immunocytochemistry**. Glass coverslips were coated with 0.1 mg/ml Poly-D-Lysine (PDL), washed, and coated with mouse Laminin 2 µg/ml (Millipore). Cells were plated on coated coverslips for 24 h, at which time they were fixed with 4% PFA and then cryoprotected using 4% sucrose. Immunocytochemistry was performed by incubating fixed cells with anti-βIII Tubulin (1:1000, Tuj1, Mouse, Promega G7121), pSTAT3 (1:1000, Rabbit, Cell Signaling Technology 9145), and p-PKC (1:1000, Rabbit, Cell Signaling Technology 9371) antibodies at 4 °C O/N. This was followed by incubation with Alexa Fluor-conjugated goat secondary antibodies according to standard protocol (1:1000, Invitrogen). Slides were counterstained with DAPI to visualize nuclei (1:5000, Molecular Probes).

**Image analysis for IHC and ICC**. All analysis was performed by the same experimenter who was blinded to the experimental groups. DRG photomicrographs were taken with a Nikon Eclipse TE2000 microscope with an opti-MOS scMOS camera using 10× or 20× magnification using ImageJ (Fiji 64bits 1.52p), Macromanager 1.4 software for image acquisition.

**Analysis of neurite outgrowth**. The mean neurite outgrowth per DRG neuron was manually measured using the NeuronJ plugin for ImageJ software (ImageJ 1.52p). We measured the mean neurite outgrowth of ~20 neurons per replicate with at least three mice per group and three technical replicates per mouse.

**Analysis of CTB-positive dorsal column axon regeneration in the spinal cord**. Regeneration of dorsal column axons were quantified from sagittal spinal cord sections from one series of tissue for each mouse. CTB intensity was quantified

using ImageJ software at set distances rostral to injury center then expressed as a percentage of the CTB intensity caudal to the injury site to control for variations in tracing efficacy.

The furthest rostral CTB-positive axon was determined relative to the lesion center and caudal axon retraction was defined as distance the lesioned axon bundle retracted relative to the lesion center.

Coronal spinal cord sections at T3 spinal level were examined to check for spared fibers above the lesion; any animals with sparing were removed from the analysis. All analyses were performed blind to the experimental group.

**Analysis of GFAP intensity around the lesion site**. GFAP intensity and area was quantified from sagittal spinal cord sections from one series of tissue for each animal. Quantification was done using ImageJ, the background was subtracted, and then the mean pixel intensity and area of immune reactivity was measured.

**Analysis of fluorescence intensity in DRG neurons**. For quantitative analysis of pixel intensity (pPKC, pSTAT3, H3K27ac, Hydrocy3, gp91phox, p47phox, and p-p47-phox), the nucleus or soma of DRG neurons were manually outlined in images from one series of stained tissue for each mouse. To minimize variability between images, pixel intensity was normalized to an unstained area and the exposure time and microscope setting were fixed throughout the acquisition.

**Analysis of vGlut1 and VGAT immunohistochemistry in proximity to motor neurons**. VGlut1 and VGAT synaptic boutons were imaged with a SP8 Leica confocal microscope. Z-stack images were taken with an average thickness of 15 μm with a step size of 0.3 μm. Sequential line scanning was performed when more than two channels were acquired. Multi-fluorescent orthogonal 3D image analysis and visualization were performed using Leica LAS X software. The Average number of vGlut1 or VGAT boutons opposed to motor neurons in the ventral horn of L1-3 spinal sections was calculated by analyzing 20 motor neurons per replicate. All analyses were performed blind to the experimental group.

**RNA sequencing**. RNAseq was performed using RNA from DRG in three biological replicates. For DRG RNAseq, sciatic DRG were extracted bilaterally from two mice that were pooled in one biological replicate and stored in RNase later (Qiagen). DRG tissue was crushed with RNAse free micropestle and RNA was extracted using RNAeasy kit (Qiagen), according to the manufacturer's protocol. Residual DNA contamination was removed by on column DNase I treatment (Qiagen) for 15 min at RT. RNA concentrations and quality were measured using Agilent 2100 Bioanalyzer (Agilent). RNA with RNA integrity number (RIN) factor above 7.5 was used for library preparation. Libraries were prepared at Ospedale San Raffaele (Milan) using the TruSeq mRNA Sample Preparation kit (Illumina) and sequenced using Illumina HiSeq 2500 100-cycle, paired-end sequencing. Sequence reads were aligned to the mm10 mouse reference genome sequence using tophat version 2.0.12 running Bowtie2-2.2.3. Gene structure annotations corresponding to the Ensembl annotation of the mm10 genome sequence were used to build a transcriptome index and provided to tophat during the alignment step. The aligned reads were sorted using samtools-0.1.19 and read counts per gene were obtained from mapped reads using HTSeq- 0.6.1. EdgeR version 3.8.6 (using limma-3.22.7) in R-3.1.1 was used to identify DE genes ($P$-value $< 0.05$). Read-level quality checking was performed using fastqc-0.10.1 and the fastqc-aggregator (https://github.com/staciaw/fastqc_aggregator), and gene-level quality checking was performed using RSeQC-2.6.1. GO was performed on the DE genes with DAVID (Database for Annotation, Visualization, and Integrated Discovery version 6.7 (http://david.abcc.ncifcrf.gov/) using a threshold of $P < 0.05$ and all the expressed genes in the RNAseq dataset as reference genome. Functional GO clustering was performed using ClueGO in Cytoscape (http://www.cytoscape.org): (Bonferroni $P > 0.05$) (GO tree interval min3, max8) (GO selection min 4, 1%) (Kscore 0.3). DE genes that resulted enriched in the GO functional clustering were uploaded into STRING 10.5 to build a protein–protein interaction network that was visualized by Cytoscape. DE gene sets for each condition were visualized by BioVenn (C) 2007–2020 Tim Hulsen. Additional four random networks were generated as control, randomly selecting, by using "rand" function in Excel, the same number of genes from the DE gene list. Connectivity parameters were calculated and statistically compared between the selected and the four randomly generated networks (Supplementary Data 5).

**P-SCAN analysis**. TF motif enrichment analysis was performed using PScan[35] on the TSS ($+50$ bp/$-950$ bp) regions of genes belonging to NOX pathway that were upregulated after EE + SNA.

**RNA extraction and cDNA synthesis**. DRG were dissected as previously described (refer to DRG extraction section) and were collected in an Eppendorf tube containing 200 μL/tube of RNA-Later solution. Tubes were left at $-20$ °C overnight and the day after tissue was crushed with a pestle and lysed by adding 200 μL/tube of freshly prepared lysis buffer (RNeasy Protect Mini Kit, Qiagen) and mercaptoethanol. RNA was extracted following standard procedure using the RNeasy Protect Mini protocol (Qiagen) including DNase I digestion step. Total RNA extracted was converted to first-strand cDNA using the SuperScript III First-

Strand Synthesis System kit (Thermo Fisher Scientific). For HEK-293 cell experiment, cells were lysed by adding 600 μL/well of freshly prepared lysis buffer (RNeasy Protect Mini Kit, Qiagen) and mercaptoethanol. Cell lysates were transferred to Eppendorf tubes and RNA was extracted following standard procedure using the RNeasy Protect Mini protocol (Qiagen) including DNase I digestion step. Total RNA extracted was converted to first-strand cDNA using the SuperScript® III First-Strand Synthesis System kit (Thermo Fisher Scientific).

**PCR and qPCR**. Taq DNA Polymerase (New England Biolabs) was used in the PCR amplification of SD STAT3 from the pCDNA3-S727D-STAT3 vector plasmid. The PCR was performed on a Mastercycler Pro (Eppendorf, Hamburg, Germany). The thermal cycle profile consisted of 1 cycle (5 min at 94 °C; 2 min at 52 °C; 7 min at 72 °C) and a further 40 cycles (1 min at 94 °C; 1 min 54 °C; 1 min 72 °C). All qPCR was carried out using SYBR Green I (2×) contained within the Platinum SYBR Green qPCR SuperMix-UDG kit (Thermo Fisher Scientific). Reactions were performed in triplicates and analyzed on a 96-well plate in a final PCR reaction volume of 20 μL using 7900HT Fast Real-Time PCR System (Applied Biosystems, Foster City, CA USA). The thermal cycle profile consisted of 45 cycles composed of 4 stages (5 min at 95 °C, 20 s at 95 °C, 20 s at 56 °C, 20 s at 72 °C) and a final dissociation stage. The threshold cycle (Ct) data after each run was exported from SDS v2.4. The Ct data for the precipitated DNA in the anti-pSTAT3 ChIP was first normalized against the 5% input. The normalized precipitated DNA values were expressed as fold change relative to the negative control, IgG. The Ct data derived from the RT-qPCR on the total RNA were first normalized to the endogenous reference gene GAPDH and expressed as fold change relative to the control (Sham for DRG and eM-EGFP for HEK cell experiments, respectively). qPCR optimization against a sixfold dilution of mouse genomic DNA was carried out prior to PCR run to determine optimal primer/DNA concentrations and annealing temperature.

**Chromatin immunoprecipitation**. A freshly prepared solution of 1% formaldehyde (Merck #F8775) was added to crushed sciatic DRG for in vivo cross-linking (15 min at RT) of the protein–DNA complexes. Crosslinking was quenched by addition of 2.5 M glycine for 5 min and centrifugation ($2500 \times g$, 3 min, 4 °C). The pellet was washed four times with PBS + protease inhibitor (Merck KGAa #11873580001) and resuspended in Lysis Buffer 1 (1 M Hepes-KOH, 5 M NaCl, 0.5 M EDTA, 50% glycerol, 10% NP-40, 10% Triton X-100). The lysed cells were rocked at 4 °C for 10 min and centrifuged ($2000 \times g$, 5 min, 4 °C). Pellet resuspended in Lysis Buffer 2 (Tris-HCl, 5 M NaCl, 0.5 M EDTA, 0.5 M EGTA). Nuclei collected by centrifugation ($2000 \times g$, 5 min, 4 °C) and resuspended in Lysis Buffer 3 (Tris-HCl, 5 M NaCl, 0.5 M EDTA, 0.5 M EGTA, 10% Na-Deoxycholate, 20% N-lauroylsarcosine). The nuclear DNA was sonicated with Bioruptor (Diagenode, Liege, Belgium) to an average length of 200–800 bp. Dynabeads Protein G (Invitrogen #10004D, Carlsbad, CA USA) were washed in blocking solution (1× PBS, 0.5% BSA, ddH2O) three times. Input was 5% of nuclear DNA samples and stored at $-80$ °C. Beads were resuspended in blocking solution and anti-pSTAT3 (Cell Signaling Technology #9145, Leiden, The Netherlands) or nonreactive rabbit IgG (negative control) and left to incubate for 6 h on a rotating platform at 4 °C. A final PBS wash is carried out to remove any excess antibody. For preclearing of the nuclear DNA, the solution was incubated with washed Dynabeads for 6 h. After incubation, the precleared DNA was added to the pre-blocked antibody/Dynabead solution and incubated overnight on a rotating platform at 4 °C. The DNA/antibody/Dynabead solution was washed five times in Wash Buffer (1 M Hepes-KOH pH 7.6, 5 M LiCl, 0.5 M EDTA, 10% NP-40, 10% Na-Deoxycholate, ddH2O) and one time in TBS (20 mM Tris pH 7.6, 150 mM NaCl). Elution buffer was added to DNA/antibody/Dynabead samples and the input samples as well, and left to incubate at 65 °C O/N for crosslink reversal. The DNA/antibody solution was separated from the Dynabeads and purified using the QIAquick PCR Purification Kit (Qiagen).

**Oligonucleotides**. All primers were supplied by Eurofins Genomics (Ebersberg, Germany). PCR based cloning of SD STAT3 2 primers, stat3F (5′-CCCAAGCTTGGGAAGCTTGATGGCTCAGTGGAACCAGCTG-3′) and stat3R (5′-GCCCCATGGGGCGGTACCGTTATAGAATAGGGCCCTTGTC3′) were designed for the PCR amplification of SD STAT3 from the pCDNA3-S727D-STAT3 vector plasmid. Stat3F was designed for amplification of the sense strand of SD STAT3 sequence and introduction of a HindIII restriction site at the 5′-end. Stat3R was designed for amplification of the nonsense strand of SD STAT3 and introduction of a KpnI restriction site at the 5′-end. Sanger sequencing of STAT3 plasmid constructs 7 forward primers were designed for Sanger sequencing of the pEGFP-WT/YF/SD STAT3 plasmid constructs. One primer each was designed for sequencing of the WT and YF-STAT3 plasmid constructs and five primers for the SD STAT3 plasmid.

stat3wt: 5′-GTAAGACCCAGATCCAGTCC-3′
stat3yf: 5′-GTAAGACCCAGATCCAGTCC-3′
stat3sd1: 5′-GCAAGAGTCCAATGTCCTCT-3′
stat3sd2: 5′-GCAATGGAGTACGTGCAGAAG-3′
stat3sd3: 5′-AGACTCTGGGGATGTTGCTG-3′
stat3sd4: 5′-TCAGGGTGTCAGATCACATG-3′
stat3sd5: 5′-AAGGAGGAGGCATTTGGAAAG-3′

**Primers**. Six primer pairs were designed for the qPCR analysis of the RNA isolated from DRG tissue. The mRNA templates for the 6 NOX2 subunit coding sequences (CDS) were derived from NCBI nucleotides database (NCBI-website). Primers were designed on the sequence spanning an intron and an exon, when possible, in order to avoid non-specific amplification of the respective gene. All primers produce a 150–400 bp PCR product. NCBI Primer-BLAST (https://www.ncbi.nlm.nih.gov/tools/primer-blast/) was used to ensure that primers were optimally designed with an appropriate balance between template specificity and thermodynamic stability.

gp91phox: sense 5′-GAGTCACGCCCTTTGCCTCC-3′, antisense 5′-CTCAAAGGCATGCGTGTCCC-3′

p22phox: sense 5′-ACCTGACCTCTGTGGTGAAGC-3′, antisense 5′-GGCAATGGCCAAGCAGACG-3′

p47phox: sense 5′-CAACAGCGTCCGATTCCTGC-3′, antisense 5′-TCGATGGATTGTCCTTTGTGCC-3′

p67phox: sense 5′-GGAACTGAAGCTCAGCGTGC-3′, antisense 5′-ACCCACCGTATGCTCACACC-3′

Rac1: sense 5′-TTTTCCCCAGCTTTGGGTGG-3′, antisense 5′-GTGTCTCCAACTGTCTGCGG-3′

NOX3: sense 5′-AGCTCTGTAGCATGCCGAGG-3′, antisense 5′-TGGGTCGCCCATAGAAAGCC-3′

Jun: sense 5′-AGCGCCTGATCATCCAGTCC-3′, antisense 5′-TCGTCGGTCACGTTCTTGGG-3′

Six primer pairs were designed for qPCR analysis of the DNA isolated from the anti-pSTAT3 ChIP. The DNA templates for the six subunit genes that constitute NOX2 were derived from the Mouse Genome Assembly GRCm38.p6 using Ensembl (https://www.ensembl.org/). Primers were designed upstream of the first exon sequence that is not part of the CDS (5′-untranslated region). All primers produce a 150–400 bp PCR product. NCBI Primer-BLAST (https://www.ncbi.nlm.nih.gov/tools/primer-blast/) was used to ensure that primers were optimally designed with an appropriate balance between template specificity and thermodynamic stability.

CYBA: sense 5′-AAGTGACTTACGCCTCTGGC-3′, antisense 5′-TTGAGGTCCGCAGAACCAGC-3′

CYBB: sense 5′-CTTATAACACTTCATCCAGGG-3′, antisense 5′-CCATTCAGCACACCATCATCG-3′

NCF1: sense 5′-CATGGTGTCGGTAGAACAGG-3′, antisense 5′-TCCTGCAGGGCACTCTTAGG-3′

NCF2: sense 5′-TTCAGCACTCTCCCGGAGG-3′, antisense 5′-TGTCAGGATGCAGGGGACC-3′

NCF4: sense 5′-CACCAGAGACAGGCACTGAG-3′, antisense 5′-CCTCACACGTCTTCTCAGCC-3′

Six primer pairs were designed for the qPCR analysis of RNA transcripts extracted from the HEK-293 cells transfected with a DNA plasmid expressing STAT3. The mRNA templates for the six subunit genes that constitute NOX2 were derived from the RefSeq human reference genome assembly using the NCBI Nucleotide Database (https://www.ncbi.nlm.nih.gov/nuccore/). The primers are designed to span an exon-exon junction and produce a 150–400 bp PCR product. NCBI Primer-BLAST was used to ensure that primers were optimally designed with an appropriate balance between template specificity and thermodynamic stability.

CYBA: sense 5′-CCAGTGGTACTTTGGTGCC-3′, antisense 5′-CACAGCCGCCAGTAGGTAG-3′

CYBB: sense 5′-CTTCACTCTGCGATTCACACC-3′, antisense 5′-AGACCTCCGGATGGTTTTGG-3′

NCF1: sense 5′-GTCCTGACGAGACGGAAGAC-3′, antisense 5′-CGTCTTTCCTGATGACCCAC-3′

NCF2: sense 5′-ATGTTCAACGGGCAGAAGGG-3′, antisense 5′-CCAGGGGCTTTGGAACTAGG-3′

NCF4: sense 5′-CGGAAAGTCAAGAGCGTGTC-3′, antisense 5′-TCGTAGGGATGGCAGGAAGG-3′

RAC1: sense 5′-AGCCGATTGCCGATGTGTTC-3′, antisense 5′-GACCCTGCGGATAGGTGATG-3′

**Behavioral assessment of sensorimotor function**. Mice were trained daily for 1 week pre-surgery before baseline measurements and then assessed on day 3 post-surgery and weekly thereafter. All behavioral testing and analysis were done by two independent observers blinded to the experimental groups.

**Grid walk**. The grid walk consists of a 50 cm × 5 cm plastic grid placed between two vertical 40 cm high wood blocks. The mesh is formed by 1 cm × 1 cm spaces. Mice were allowed to run the grid walk three times per session. The number of total steps and missteps per run for each hind paw were analyzed by a blind investigator.

**Adhesive removal**. An 8 mm diameter adhesive pad was placed on each hind paw. The mouse was then placed into plexi-glass box and the time until it first contacted each of the adhesive pads was recorded, followed by the time until it removed the adhesive pads from each hind paw. The maximum time allowed for each animal was 5 min. Each animal was tested twice per time point and values represent the average time from both hind paws from both runs.

**Statistical analysis**. Results are expressed as mean ± SEM. Statistical analysis was carried out using GraphPad Prism 8. Normality was tested for using the Shapiro–Wilk test. Normally distributed data were evaluated using a two-tailed unpaired Student's $t$-test or a one-way ANOVA when experiments contained more than two groups. The Tukey's post hoc test was applied when appropriate. The Mann–Whitney's $U$-test was used for non-parametric evaluation. A threshold level of significance α was set at $P$-value < 0.05. Significance levels were defined as follows: *$P$-value < 0.05; **$P$-value < 0.01; ***$P$-value < 0.001, ****$P$-value < 0.0001. All data analysis was performed blind to the experimental group.

**Reporting summary**. Further information on experimental design is available in the Nature Research Reporting Summary linked to this paper.

## Data availability
The RNAseq data have been deposited in GEO with the following number: GSE138769. The mass spectrometry proteomics data have been deposited to the ProteomeXchange Consortium via the PRIDE partner repository67 with the dataset identifier PXD018085. Source data are provided with this paper.

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

## Acknowledgements

We thank Mike Fainzilber (M.F.) (Weizmann Institute) for critical discussion. We thank Jana Meisterknecht for excellent technical assistance and sample preparation for mass spectrometry. This work was supported by grants from Wings for Life (S.D.G. and T.H. H.); the Rosetrees Trust (S.D.G.), Leverhulme Trust (S.D.G.), the MRC (S.D.G), the Weizmann-UK (S.D.G. and M.F.); start-up funds from the Department of Brain Sciences, Imperial College London (S.D.G.); The Miami Project to Cure Paralysis (V.P.L. and J.L.B.); The Walter G. Ross Foundation (V.P.L. and J.L.B.); the National Institute for Health (NIH) R01 HD057632 (V.P.L. and J.L.B.), The British Heart Foundation (CH/ 1999001/11735) (A.M.S.), and by the Deutsche Forschungsgemeinschaft (DFG, SFB 815/ Z1) (I.W.).

## Author contributions

F.D.V. and T.H.H contributed equally to this work. F.D.V., T.H.H., I.P., and S.D.G. conceived and designed the studies. F.D.V., T.H.H., I.W., and S.D.G. supervised the studies. F.D.V., T.H.H., I.P., S.A., J.M., L.Z., R. Todorova and I.W. performed experiments, and collected and analyzed data. M.C.D. performed data analysis. R. Thompson provided technical assistance. F.D.V. and T.H.H. prepared the figures. F.D.V., T.H.H., and S.D.G. wrote the manuscript. I.P., V.P.L, J.L.B., I.W., and A.M.S. edited the manuscript.

## Funding

## Competing interests

The authors declare no competing interests.
