## [Peer Review File · Nature Communications]

REVIEWER COMMENTS

Reviewer #1 (Remarks to the Author):

De Virgillis et al. are following up on a recent publication from their lab in which they found that environmental enrichment (EE) enhances the regenerative capacity of dorsal root ganglion (DRG) axons after a spinal cord injury (SCI). In the series of studies described in this manuscript, they test the hypothesis that performing a conditioning sciatic nerve axotomy (SNA) in animals that have EE, a combination that they call enriched conditioning, will have an additive effect on axon regeneration. They report that enriched conditioning improves sensory axon growth in a manner that is dependent upon PKC-STAT3-NOX2. Lastly, they report that driving NOX2 signaling is sufficient to enhance growth after a dorsal columns SCI and spinal circuit reorganization below the level of injury that correlates with improved performance in a tape removal task.

This is a well-written manuscript that described a logically designed set of experiments. The unbiased proteomics approach to identify candidate mechanisms that were then validated is a strength. While environmental enrichment and a conditioning lesion are not novel interventions to promote axon growth, the combination is. Additionally, while PKC, STAT3, and NOX2 have all been implicated in enhancing growth, the identification of them being interrelated is novel. Additional novelty lies in the identification of a mechanism that appears to play a larger role in the combination than either intervention singly.

There are a few concerns:

1. Would the authors please comment on why they chose a 24hr post-SNA time point? Conditioning lesion effects are usually examined several days after the injury.
2. In Fig. 6C, it is not clear if the asterisks indicate that EE SNA is significantly different from EE SNA-STAT3CKO, SH Sham, or both of the other groups. Additionally, the authors indicate that there is no significant difference in axon between EE SNA+STAT3CKO and the sham controls, though the bars appear to be quite distinct. It would be helpful to indicate what the p value is.
3. The authors report that overexpression of the constitutively active p47phox induces axon regeneration. Is this effect comparable to the effect seen with enriched conditioning?
4. The authors injected the AAV into the sciatic. Does this transduce non-neuronal cells, which may affect the growth capacity of neurons indirectly? Have they examined the extent of GFP expression at the injection site?
5. The authors found that there was an increase in tape removal ability in the p47phox animals with a dorsal columns SCI, which correlated with more vGAT and vGlut1 positive boutons in surrounding lumbar motor neurons. The authors appear to conclude that synaptic reorganization associated with growth mediates the functional recovery. However, it is not clear how much of this recovery is actually associated with sensory axon growth, per se, since the SCI was more rostral in thoracic spinal cord, there is no evidence that the axon regeneration in the dorsal columns is reaching the dorsal columns nuclei in the brain stem, and there is no data presented indicating growth of sensory axons in lumbar spinal cord (below the level of SCI). Is there sprouting of GFP positive fibers in lumbar cord? Or is the increase in vGAT and vGlut1 boutons reflective of synaptic plasticity in the absence of axon growth? These could be completely independent mechanisms. The desire to demonstrate functional recovery is appreciated but without more data on growth, these functional data somewhat detract from the sharp focus of the rest of the manuscript.

Reviewer #2 (Remarks to the Author):

In this study, the authors present evidence that combining conditioning lesion (sciatic nerve axotomy) with enriched environment prior to a spinal cord injury promotes axonal regeneration of DRG neurons beyond the spinal cord injury site to a larger extent than conditioning lesion alone. The authors called this experimental paradigm enriched conditioning and they show that enriched conditioning prior spinal cord injury promotes axonal regeneration through NADPH oxidase 2 (NOX2), which they find upregulated by enriched conditioning after spinal cord injury. The authors show that NOX2 is upregulated by phospho-Stat3, the latter being phosphorylated by PKC after injury. NOX2 produces reactive oxygen species (ROS), thus this study suggests that the production of ROS and the activation of some specific redox signaling pathways is in this case beneficial to enhance axonal regeneration after lesion. Analysis of oxidized proteins identifies proteins related to response to injury and wound healing, signaling, lipid metabolism, extracellular matrix and cell adhesion. How oxidation of these proteins promotes axonal regeneration remains to be elucidated.

This study is very interesting, in general very well conducted with clear aims and conclusions, and identifies a potential strategy to follow up to promote axonal regeneration of central neurons. I have however a few comments that need to be addressed before publication:

1/ Page 6, 1st paragraph: « Importantly, the elevation in hydrocyanine-Cy3 signal was blocked in DRG from mice where NOX2 (gp91phoxfl/fl) was conditionally deleted at all the time points analyzed (Figure 4A and B and Supplementary Figure 8).

There is a mistake in the number of the Supplementary Figure cited. It should be Supplementary Figure 7 instead of 8.

2/ Supplementary Figure 10: Are the authors sure that the error bars are SEM? They seem more like standard deviations.

3/ Could it be ruled out that the AAV injected in the sciatic nerve infect Schwann cells and that this has an impact on the results?

4/ Figure 6a-c: It is not clear here whether EE-SNA has been transduced with AAV-GFP to control EE-SNA-Stat3KO that has been injected with AAV-Cre-GFP (such as shown in Supplementary Fig. 11). It is essential that the control is injected with a control AAV. If it is the case, please mention it in the figure and figure legend. If it is not the case, this experiment needs to be redone with the appropriate control.

5/ Figure 6d-f: It is also not clear here whether the controls SH Sham and EE-SNA were treated with the vehicle of the PKC activator and the PKC inhibitor, respectively. If they were, please add this information to the figures and figure legends. If they were not, please re-do the experiments with the appropriate controls.

6/ Figure 7e-h: It is not clear what n is. Is it the number of motoneurons or the number of animals. If it is 9 motoneurons in a single animals, it is not good enough. Individual values should be the average of several motoneurons (precise how many) per animal and n should be at least 3 animals per group. Please, indicate what n is in the figure legend and how many motoneurons per animals were quantified. If necessary, add more animals.

7/ Figure 7e-g: Are these images confocal images? Confocal images are necessary here to determine whether vGAT and vGlut1-positive boutons are in contact with motoneurons.

Reviewer #3 (Remarks to the Author):

This paper by De Virgiliis and colleagues explores the mechanisms by which an enriched environment significantly enhances, in an additive manner with sciatic nerve axotomy, regeneration from spinal cord injury.

This builds on the group's previous work on NOX2.

The authors first show that NOX-dependent REDOX signaling is selectively affected by the combination of EE+SNA in combination using RNAseq data. Following confirmation that NOX2 subunits are increased, they then use a series of Cre-knock-out animals to show the importance of NOX2 mechanisms in the regeneration process, including deletion of gp91phox and later overexpression of the constitutively active p47phox.

Overall, I think this is an interesting paper with good evidence that Nox2 plays an important role in EE+SNA enhanced regeneration after SCI.

I am not sure that the full mechanistic pathway has been robustly delineated i.e. how EE+SNA together lead to altered NOX2 and how altered NOX2 enhances regeneration - but the work remains impactful nevertheless.

There is a section on oxidised proteins, which is interesting but as described presently, does not really fit into the narrative. In the list of oxidised proteins, do any show overlap with genes implicated in nerve regeneration or outgrowth - the authors mention that cell signaling and wound healing are terms identified but what are the specific genes?; what is the functional consequence of oxidation of these proteins? - are they targeted for degradation?

The quantification of regeneration, particularly the %CTB intensity rostral/caudal which is the main measure used to assess regeneration after SCI in the various models, seems to be quite variable. Hence in Figure 3E, the % at 0-200uM is ~40% compared; whilst Figure 6C, the STAT3 KO (which is proposed to reduce regeneration) is also around 30-40% and Figure 7c, the overexpression of p47phox (which is proposed to increase regeneration) is also at 30-40%. The baseline (e.g. SH Sham) is fairly consistent at 10% in all these experiments.

So overall whilst I am persuaded that NOX2 does have a role in EE enhanced regeneration, I think it is likely that other pathways also operate and it would be appreciated if the authors could appropriately nuance the abstract and the discussion accordingly.

My additional comments are:

1. Were the analysis of images performed blinded to the conditions that the animals were exposed to?

2. The RNAseq analysis - do the samples cluster on a PCA plot with separation along axis according to the treatment?

3. The NOX functional annotation on Figure 2 has a ?just-significant p-value - how many members of this group were significantly differentially expressed?

4. Figure 2C illustrates a network centred around NOX proteins - is it possible to provide some quantitative measure of the connectivity for each gene/protein (e.g. number of edges to the other genes that are differentially expressed)?...if a random set of genes were selected, what would the average connectivity be?

5. Figure 2C 'proteins belonging to signaling pathways specifically enriched upon EE+SNA' - were these only genes showing differential expression or was it any gene in the pathway irrespective of differential expression. The former is

the appropriate figure.

6. Figure 2D - FC are shown for the NOX2 but are these significantly differentially expressed?

7. All the figures with DRG cultures, the legend states analysis was e.g. n=4/group...please provide further details on how many cells were analysed per culture?...I assume it was more than 4 cells..

8. Figure 6C - There is a clear separation so I am surprised there was no statistically significant separation between SH SHAM and STAT3KO on %CTB - what was the p-value; as noted above the % CTB is in a similar range to the over-expression experiment later in the manuscript; in this case the n is slightly lower. I also appreciate the curves only diverge at two measurement points.

I think this is not a strong figure panel and I think needs additional n to decide one way or another..if there remains significant regeneration, this would not detract overall but just means STAT3 is one of several TFs that might be involved in the EE effect.

9. The PKC is potentially a key component of the EE mechanism but the evidence presented in the manuscript is relatively weak. It would be strengthened by directly measuring PKC activity with a kinase assay in DRG after SH, EE, SH+SNA, EE+SNA

We would like to thank the reviewers for their support of our work and the constructive points they raised.

REVIEWER COMMENTS

Reviewer #1 (Remarks to the Author):

De Virgillis et al. are following up on a recent publication from their lab in which they found that environmental enrichment (EE) enhances the regenerative capacity of dorsal root ganglion (DRG) axons after a spinal cord injury (SCI). In the series of studies described in this manuscript, they test the hypothesis that performing a conditioning sciatic nerve axotomy (SNA) in animals that have EE, a combination that they call enriched conditioning, will have an additive effect on axon regeneration. They report that enriched conditioning improves sensory axon growth in a manner that is dependent upon PKC-STAT3-NOX2. Lastly, they report that driving NOX2 signaling is sufficient to enhance growth after a dorsal columns SCI and spinal circuit reorganization below the level of injury that correlates with improved performance in a tape removal task.

This is a well-written manuscript that described a logically designed set of experiments. The unbiased proteomics approach to identify candidate mechanisms that were then validated is a strength. While environmental enrichment and a conditioning lesion are not novel interventions to promote axon growth, the combination is. Additionally, while PKC, STAT3, and NOX2 have all been implicated in enhancing growth, the identification of them being interrelated is novel. Additional novelty lies in the identification of a mechanism that appears to play a larger role in the combination than either intervention singly.

There are a few concerns:

We would like to thank the reviewer for their support of the manuscript and their suggestions for improving it.

1. Would the authors please comment on why they chose a 24hr post-SNA time point? Conditioning lesion effects are usually examined several days after the injury.

We thank the reviewer for giving us the opportunity to discuss this point. Other studies often use a 7-day timepoint when studying the conditioning lesion paradigm. However, Smith and Skene. J Neurosci. 1997 showed that a conditioning injury performed 24hr before culturing DRG neurons is sufficient to increase the percentage of DRGs that had transitioned to an axon elongating phenotype. Furthermore, the axon regeneration that we report in the spinal cord 24 hours after a conditioning injury (0.6mm from injury site) is similar to what is reported by Neumann and Woolf. J Neuron. 1999 (approx. 0.5mm from the injury site) after they performed a conditioning injury 7 days before the SCI. Therefore, we have found that 24hr post conditioning injury is sufficient to observe an increase in both neurite outgrowth and axon regeneration and we have previously used this method in several publications (Hervera et al Nat Cell Biol 2018, Hutson et al. Sci Transl Med 2019). Importantly we also chose the 24hr time point so that we could detect early gene expression change underpinning the phenotype in addition to allowing us to compare our RNAseq dataset with several other transcriptomic studies that are performed at the 24hour time point, including ours. The 24hr is commonly used to avoid the large macrophage infiltration into the DRG that occurs at later timepoints, which could heavily affect the RNAseq analysis.

2. In Fig. 6C, it is not clear if the asterisks indicate that EE SNA is significantly different from EE SNA-

STAT3CKO, SH Sham, or both of the other groups. Additionally, the authors indicate that there is no significant difference in axon between EE SNA+STAT3CKO and the sham controls, though the bars appear to be quite distinct. It would be helpful to indicate what the p value is.

Figure labeling was not optimal here and this led to some confusion, apologies. EESNA is significantly different from both groups. P-values have been added in Fig 6C accordingly and this clarifies the issue now.

3. The authors report that overexpression of the constitutively active p47phox induces axon regeneration. Is this effect comparable to the effect seen with enriched conditioning?

Thank you for giving us the chance to discuss this point. Caution must be used when comparing regeneration values between different experiments due to inherent variability associated with performing experiments at different times. However, if we do compare between experiments the percentage intensity of regenerative fibres at 0, 200 and 400um rostral to the injury site are very similar in both conditions (40% at 0, 30% at 200um and 20% at 400um). At distances further from the lesion site there is more variability between animals. However, the enriched conditioning paradigm generally promotes more regeneration than p47-3X (EE+SNA vs p47-3X: 23% vs 12% at 600um, 14% vs 6% at 800um and 7% vs 2% at 1000um). This suggests that other signalling pathways and mechanisms may also contribute to the regeneration observed after enriched conditioning. This is not unexpected, and we hope that the enriched conditioning model will aid the discovery of novel targets for axonal regeneration in future studies.

4. The authors injected the AAV into the sciatic. Does this transduce non-neuronal cells, which may affect the growth capacity of neurons indirectly? Have they examined the extent of GFP expression at the injection site?

We thank the reviewer for this comment and agree it is a very interesting question. We have analysed sciatic nerve injection sites and found very little GFP expression co-localised with the Schwann cell marker SOX10 or monocyte/macrophage marker CD68 (suppl. fig. 16). However, we observed approx. 80% co-localisation of GFP with the axonal marker TUJ1. This suggests that AAV8 preferentially transduces neurons and not Schwann cells or monocytes/macrophages after injection in the sciatic nerve.

5. The authors found that there was an increase in tape removal ability in the p47phox animals with a dorsal columns SCI, which correlated with more vGAT and vGlut1 positive boutons in surrounding lumbar motor neurons. The authors appear to conclude that synaptic reorganization associated with growth mediates the functional recovery. However, it is not clear how much of this recovery is actually associated with sensory axon growth, per se, since the SCI was more rostral in thoracic spinal cord, there is no evidence that the axon regeneration in the dorsal columns is reaching the dorsal columns nuclei in the brain stem, and there is no data presented indicating growth of sensory axons in lumbar spinal cord (below the level of SCI). Is there sprouting of GFP positive fibers in lumbar cord? Or is the increase in vGAT and vGlut1 boutons reflective of synaptic plasticity in the absence of axon growth? These could be completely independent mechanisms. The desire to demonstrate functional recovery is appreciated but without more data on growth, these functional data somewhat detract from the sharp focus of the rest of the manuscript.

Thank you for giving us the chance to discuss and clarify this point. We did not observe long-distance axon regeneration reaching the brainstem nuclei nor increased sprouting of sensory axons below the injury. Our data on vGAT and vGlut1 boutons suggest that there is spinal circuit reorganization and synaptic plasticity below the injury site between motor neurons and both inhibitory propriospinal neurons (vGAT) and excitatory group-Ia proprioceptive afferents (vGlut1). NOX2 has previously been shown to play a crucial role in the induction of LTP and synaptic plasticity in the primary visual cortex and hippocampus of the mouse (Kishida, KT et al. Mol Cell Biol. 2006; De Pasquale, R et al. J Neurosci. 2014). Furthermore, we believe this synaptic reorganisation could underlie the improvements in

functional recovery as this type of plasticity has previously been associated with recovery of function in animals and humans (Takeoka, A et al. Cell 2014; van den Brand, R. et al. Science 2012; Darian-Smith, C. Neuroscientist 2009).

Reviewer #2 (Remarks to the Author):

In this study, the authors present evidence that combining conditioning lesion (sciatic nerve axotomy) with enriched environment prior to a spinal cord injury promotes axonal regeneration of DRG neurons beyond the spinal cord injury site to a larger extent than conditioning lesion alone. The authors called this experimental paradigm enriched conditioning and they show that enriched conditioning prior spinal cord injury promotes axonal regeneration through NADPH oxidase 2 (NOX2), which they find upregulated by enriched conditioning after spinal cord injury. The authors show that NOX2 is upregulated by phospho-Stat3, the latter being phosphorylated by PKC after injury. NOX2 produces reactive oxygen species (ROS), thus this study suggests that the production of ROS and the activation of some specific redox signaling pathways is in this case beneficial to enhance axonal regeneration after lesion. Analysis of oxidized proteins identifies proteins related to response to injury and wound healing, signaling, lipid metabolism, extracellular matrix and cell adhesion. How oxidation of these proteins promotes axonal regeneration remains to be elucidated.

This study is very interesting, in general very well conducted with clear aims and conclusions, and identifies a potential strategy to follow up to promote axonal regeneration of central neurons. I have however a few comments that need to be addressed before publication:

We would like to thank the reviewer for their appreciation of our work and their useful comments.

1/ Page 6, 1st paragraph: « Importantly, the elevation in hydrocyanine-Cy3 signal was blocked in DRG from mice where NOX2 (gp91phox^{fl/fl}) was conditionally deleted at all the time points analyzed (Figure 4A and B and Supplementary Figure 8).

There is a mistake in the number of the Supplementary Figure cited. It should be Supplementary Figure 7 instead of 8.

Thank you and apologies, these have now been corrected.

2/ Supplementary Figure 10: Are the authors sure that the error bars are SEM? They seem more like standard deviations.

Yes, we can confirm that they are SEM.

3/ Could it be ruled out that the AAV injected in the sciatic nerve infect Schwann cells and that this has an impact on the results?

We agree that this is a very important point. We have analysed sciatic nerve injection sites and found very little GFP expression co-localised with the Schwann cell marker SOX10 or monocyte/macrophage marker CD68 (suppl. fig. 16). However, we observed approx. 80% co-localisation of GFP with the axonal marker TUJ1. This suggests that AAV8 preferentially transduces neurons and not Schwann cells or monocytes/macrophages after injection in the sciatic nerve.

4/ Figure 6a-c: It is not clear here whether EE-SNA has been transduced with AAV-GFP to control EE-SNA-Stat3KO that has been injected with AAV-Cre-GFP (such as shown in Supplementary Fig. 11). It is essential that the control is injected with a control AAV. If it is the case, please mention it in the figure and figure legend. If it is not the case, this experiment needs to be redone with the appropriate control.

We can confirm that AAV-GFP has been used as control. The Figure and figure legends have been updated accordingly.

5/ Figure 6d-f: It is also not clear here whether the controls SH Sham and EE-SNA were treated with the vehicle of the PKC activator and the PKC inhibitor, respectively. If they were, please add this information to the figures and figure legends. If they were not, please re-do the experiments with the appropriate controls.

We can confirm that the vehicle for both drugs was DMSO and that both SH Sham and EESNA were treated with vehicle. The figure and figure legend have been updated accordingly.

6/ Figure 7e-h: It is not clear what n is. Is it the number of motoneurons or the number of animals. If it is 9 motoneurons in a single animals, it is not good enough. Individual values should be the average of several motoneurons (precise how many) per animal and n should be at least 3 animals per group. Please, indicate what n is in the figure legend and how many motoneurons per animals were quantified. If necessary, add more animals.

Thank you for giving us the chance to clarify this. The n refers to the number of animals and we analysed 15-20 motor neurons per animal. The figure and figure legends have been updated accordingly.

7/ Figure 7e-g: Are these images confocal images? Confocal images are necessary here to determine whether vGAT and vGlut1-positive boutons are in contact with motoneurons.

We agree with this suggestion. We have performed confocal imaging of synaptic boutons and included multi-fluorescent orthogonal 3D confocal images that demonstrate both vGlut1 and vGAT-positive boutons are close contact with motor-neurons. We have updated figure 7e-g and figure legends accordingly.

Reviewer #3 (Remarks to the Author):

This paper by De Virgiliis and colleagues explores the mechanisms by which an enriched environment significantly enhances, in additive manner with sciatic nerve axotomy, regeneration from spinal cord injury.

This builds on the groups previous work on NOX2.

The authors first show that NOX-dependent REDOX signaling is selectively affected by the combination of EE+SNA in combination using RNAseq data. Following confirmation that NOX2 subunits are increased, they then use a series of Cre-knock-out animals to show the importance of NOX2 mechanisms in the regeneration process, including deletion of gp91phox and later overexpression of the constitutively active p47phox.

Overall, I think this an interesting paper with good evidence that Nox2 plays an important role in EE+SNA enhanced regeneration after SCI.

I am not sure that the full mechanistic pathway has been robustly delineated i.e how EE+SNA together lead to altered NOX2 and how altered NOX2 enhances regeneration - but the work remains impactful nevertheless

We would like to thank the reviewer for their appreciation of our work and for their suggestions, which we believe have strengthened the manuscript

There is a section on oxidised proteins, which is interesting but as described presently, does not really fit into the narrative. In the list of oxidised proteins, do any show overlap with genes implication in nerve regeneration or outgrowth - the authors mention that cell signaling and wound healing are terms identified but what are the specific genes?; what is the functional consequence of oxidisation of this proteins? - are they targeted for degradation

We thank the reviewer for the important comment and we agree that protein oxidation does not imply that proteins will be degraded. This methodology identifies reversible redox modifications (e.g. redox signalling) that have been associated with cell signalling and regeneration of a number of tissues and cells, including sensory neurons as we previously reported (see for example Hervera, A et al. Nat Cell Biol. 2018). In fact, our data suggest a NOX2-dependent shift from mitochondrial respiration to redox signalling, specifically directed to cellular repair processes. However, further studies will be needed to shed light on the role of the oxidation of specific proteins identified here and their involvement in axonal regeneration processes.

The quantification of regeneration, particularly the %CTB intensity rostral/caudal which is the main measure used to assess regeneration after SCI in the various models, seems to be quite variable. Hence in Figure 3E, the % at 0-200uM is ~40% compared; whilst Figure 6C, the STAT3 KO (which is proposed to reduce regeneration) is also around 30-40% and Figure 7c, the overexpression of p47phox (which is proposed to increase regeneration) is also at 30-40%. The baseline (e.g. SH Sham) is fairly consistent at 10% in all these experiments. So overall whilst I am persuaded that NOX2 does have a role in EE enhanced regeneration, I think it is likely that other pathways also operate and it would be appreciated if the authors could appropriately nuance the abstract and the discussion accordingly.

We agree with this comment. While our study demonstrates that a novel PKC-STAT3-NOX2 signalling pathway is important for the enriched conditioning effect, we believe that there will be other transcription factors and signalling mechanisms involved in this regenerative phenotype. This is a similar situation to when a conditioning lesion was discovered and cAMP signalling was found to be critical---later several other factors have been implicated in conditioning-dependent axonal regeneration after SCI. We have added a sentence in the discussion (page 10).

My additional comments are:

1. Were the analysis of images performed blinded to the conditions that the animals were exposed to?

Yes, they were. As specified in the methods section all analysis were performed blind to experimental conditions.

2. The RNAseq analysis - do the samples cluster on a PCA plot with separation along axis according to the treatment?

We thank the reviewer for the important comment. The PCA analysis of our samples shows a clear separation between the groups as shown by the PCA plot below (rebuttal fig 2, below), which shows the separation of each biological replicate for each condition (color-coded dots) according to principal component 1 (PC1) and PC2.

3. The NOX functional annotation on Figure 2 has a just-significant p-value - how many members of this group were significantly differentially expressed?

All the genes used in the analyses are significantly differentially expressed genes (P-value <0.05)

4. Figure 2C illustrates a network centred around NOX proteins - is it possible to provide some quantitative measure of the connectivity for each gene/protein (e.g. number of edges to the other genes that are differentially expressed)?...if a random set of genes were selected, what would the average connectivity be?

We thank this reviewer for giving us the chance to clarify this. To build the network, we have uploaded into String 572 genes that were DE (P-value<0.05) in EESNA and that were significantly enriched (P-value<0.05) in GO molecular function categories related to DAG_PKC signalling, GTPase activity, phosphatase activity, NADH/NADPH dependent redox activities (categories that were enriched in EESNA with a lower P-value with respect to the other comparisons). The protein-protein network was visualized via Cytoscape and quantitative measures of the connectivity reported. Please see the network analysis in the attached Excel file for reviewer 3. In the 'network analysis sheet', connectivity parameters for our network have been included. Additionally, the 'node analysis sheet'

shows all the parameters for each node. It can be appreciated that the NOX oxidases are top ranked according to the various connectivity parameters, demonstrating the centrality of this group of proteins in our network analysis.

Furthermore, we appreciated the suggestion to compare our network to a network built with randomly selected set of genes. We randomly selected (with "rand" function in Excel) 572 genes from our DE gene list and we input them into String. The resulting network has an overall lower connectivity (larger diameter, lower centralization, lower number of neighbours), indicating the specific enriched connectivity of our network. All the parameters of the random network are reported in the attached file. In the legend sheet, the explanation of all of the parameters are reported

5. Figure 2C 'proteins belonging to signalling pathways specifically enriched upon EE+SNA' - were these only genes showing differential expression or was it any gene in the pathway irrespective of differential expression. The former is the appropriate figure.

All the genes used in the analyses are significantly differentially expressed genes (P-value <0.05)

6. Figure 2D - FC are shown for the NOX2 but are these significantly differentially expressed?

They are and all the genes used in the analyses are significantly differentially expressed genes (P-value <0.05)

7. All the figures with DRG cultures, the legend states analysis was e.g. n=4/group...please provide further details on how many cells were analysed per culture?...I assume it was more than 4 cells..
We can confirm that the n refers to the number of animals/group, for each animal approx. 20 cells were analysed. Figure legends have been corrected accordingly.

8. Figure 6C - There is a clear separation so I am surprised there was no statistically significant separation between SH SHAM and STAT3KO on %CTB - what was the p-value; as noted above the % CTB is in a similar range to the over-expression experiment later in the manuscript; in this case the n is slightly lower. I also appreciate the curves only diverge at two measurement points.

P-value has been added in Fig6C and figure legends accordingly

I think this is not a strong figure panel and I think needs additional n to decide one way or another. If there remains significant regeneration, this would not detract overall but just means STAT3 is one of several TFs that might be involved in the EE effect.

Thank you for this comment. While our data demonstrates that STAT3-dependent NOX2 expression is important for enriched conditioning-dependent axonal regeneration, we do not exclude the possible involvement of other transcription factors. For instance, table 1 shows a number of TFs that may be potentially involved, and further studies are needed to demonstrate whether these TFs play a role. We have also added a sentence in the discussion (page 10) stating this. We think this is a strength of our new enriched conditioning model that will hopefully allow the discovery of additional novel pathways and therapeutic targets to enhance axonal regeneration and improve functional recovery.

9. The PKC is potentially a key component of the EE mechanism but the evidence presented in the manuscript is relatively weak. It would be strengthened by directly measuring PKC activity with a kinase assay in DRG after SH, EE, SH+SNA, EE+SNA

Thank you for this suggestion, we agree that the addition of a PKC activity assay would strengthen the immunohistochemical and pharmacological evidence that we already provide. We did perform a PKC enzymatic activity measurement using the PepTag Non-Radioactive Protein Kinase Assay kit (Promega, V5330, USA). In this kit, a fluorescent positive-charged PKC specific peptide substrate (P-L-S-R-T-L-S-V-A-A-K) is provided. The substrate reaction solution was mixed with lysis buffer (negative control, Lane 1), recombinant PKC (positive control, Lane 2) or DRG lysate (Lane 3). The assay requires 1g of tissue in 5mL lysis buffer for tissue homogenization. For this reaction, we pooled DRG from 10 mice (30mg of DRG tissue) and homogenized them in 500uL.

In the figure above, the intensity of the bands is a measure of the non-phosphorylated substrate (Non-PY, upper band) or the phosphorylated substrate (PY, lower band). The gel shows that only the positive control, which contains recombinant PKC protein, could phosphorylate the substrate (Lane 2). No PKC activity was detected in the negative control (Lane 1) nor in the DRG lysate (Lane 3). Importantly, the intensity of the band in lane 3 (DRG lysate) is significantly lower compared to Lane 1 and 2, suggesting that the protein amount in the input lysate may be insufficient for the reaction to be detected.

Unfortunately, the assay requires a large amount of protein (1g of tissue), sciatic DRG are only 6 and very small (3mg of tissue from 1 animal), making difficult to collect 1g of tissue without sacrificing large number of animals, which raises ethical issues. Furthermore, since the DRG have an heterogenous cell population that includes non-neuronal glia and endothelial cells among others, the PKC assay should ultimately be performed from a purified neuronal preparation further multiplying the needed number of animals- making this a daunting task.

However, we believe that the evidence we provide is sufficient to argue for a role of PKC in the enriched conditioning regenerative mechanism. In fact, we show that:

1. Immunohistochemistry demonstrates that active pPKC is clearly and significantly increased in the DRG neurons after enriched conditioning (Supp Figure 14).
2. Activation of PKC using I3A significantly enhanced DRG neurite outgrowth and phosphorylation of STAT3- read out of PKC activity (Figure 6D-F).
3. Conversely, PKC inhibition with Gö 6983 significantly reduced the EE+SNA-dependent increase in neurite outgrowth and STAT3 phosphorylation (Figure 6D-F).

Taken together, these data show that *enriched conditioning* (EE+SNA) requires PKC activity to phosphorylate STAT3 and increase neurite outgrowth of DRG neurons.

REVIEWER COMMENTS

Reviewer #1 (Remarks to the Author):

This revised manuscript by DeVirgiliis et al. is much improved and the authors have addressed most of my previous concerns sufficiently. While this reviewer is still enthusiastic overall, there is still a lingering concern about the functional recovery and synaptic plasticity data presented on pages 8-9 and Figure 7E-J. This reviewer appreciates that the functional recovery observed is likely due to synaptic plasticity. One piece of evidence for this is the observed changes in vGlut1+ and vGAT1+ synaptic contacts on motoneurons. However, the analysis of vGlut1 and vGAT1 is not specific to projections from DRG neurons expressing p47-3X-GFP or GFP control. Focusing analysis on contacts made between GFP-expressing axons from neurons that have constitutively active p47phox (or the GFP control reporter) would be more relevant and would strengthen how these data fit with the rest of the manuscript.

Reviewer #2 (Remarks to the Author):

The authors have appropriately answered my comments. I therefore recommend publication.

Reviewer #3 (Remarks to the Author):

Thank you for the responses to my comments, particularly clarifying the significance values and confirming the clear separation of samples/conditions for the RNAseq analysis on the PCA plot; and the further details of the network properties.

The randomised network should ideally be repeated to then generate a significance value for the network they illustrate - but I don't think this is essential as it is not the main focus of the paper

I also note the PKC kinase activity assay is technically challenging due to the tissue requirements.

The separation between SH SHAM and STAT3KO on Figure 6C is trending to significance and I suspect with increased n, this would become clearer (in other similar experiments, the authors use n=5). What are the power calculations for these experiments...given the variance of this assay, does n=4 given sufficient power to detect differences?

We would like to thank the reviewers for their support of our work and the further constructive discussion towards our revision.

Reviewer #1 (Remarks to the Author):

This revised manuscript by DeVirgiliis et al. is much improved and the authors have addressed most of my previous concerns sufficiently. While this reviewer is still enthusiastic overall, there is still a lingering concern about the functional recovery and synaptic plasticity data presented on pages 8-9 and Figure 7E-J. This reviewer appreciates that the functional recovery observed is likely due to synaptic plasticity. One piece of evidence for this is the observed changes in vGlut1+ and vGAT1+ synaptic contacts on motoneurons. However, the analysis of vGlut1 and vGAT1 is not specific to projections from DRG neurons expressing p47-3X-GFP or GFP control. Focusing analysis on contacts made between GFP-expressing axons from neurons that have constitutively active p47phox (or the GFP control reporter) would be more relevant and would strengthen how these data fit with the rest of the manuscript.

We fully agree with this very sensible reasoning. We have now provided revised images in Figure 7G where we show co-localization between vGlut1+ puncta and GFP+ axons in contact with ChAT+ motoneurons. New quantification (Figure 7H) has also been provided to reflect this analysis that has been embedded in the previous bar graphs.

Reviewer #3 (Remarks to the Author):

Thank you for the responses to my comments, particularly clarifying the significance values and confirming the clear separation of samples/conditions for the RNAseq analysis on the PCA plot; and the further details of the network properties.

The randomised network should ideally be repeated to then generate a significance value for the network they illustrate - but I don't think this is essential as it is not the main focus of the paper

While we agree this is not an essential component of the manuscript, we have nevertheless performed a statistical analysis applied to the randomised networks since we believe it would further support our model. The analysis illustrates a statistical significance validating the selection of our gene network. We have generated 4 random networks, randomly selecting (with "rand" function in Excel) 572 genes from our DE gene list that have been uploaded into STRING. The resulting networks have a statistically significant lower connectivity (larger diameter, lower centralization, lower number of neighbours) compared to the selected network (now Supplementary File 5).

I also note the PKC kinase activity assay is technically challenging due to the tissue requirements.

The separation between SH SHAM and STAT3KO on Figure 6C is trending to significance and I suspect with increased n, this would become clearer (in other similar experiments, the authors use n=5). What are the power calculations for these experiments...given the variance of this assay, does n=4 given sufficient power to detect differences?

Thank you for this question. We performed power calculations before our experiments and the minimum number of animals allowing to show a difference in axonal regeneration between SH SHAM and STAT3KO was 4. However, we always perform surgery on additional mice due to experimental and technical variability. In this case we did surgeries and tracing in 6 mice per group,

however while only 4 had suitable tracing for inclusion in the analysis, this still met our requirement based on the power calculation. Although there was a trend to significance between SH SHAM and STAT3KO ($p=0,07$), STAT3KO still significantly reduced the *enriched conditioning* (EE+SNA) effect on axon regeneration compared to EE+SNA (Two-way repeated measures ANOVA, Tukey's post-hoc, *** P-value <0.0001). This data strongly indicates a role for STAT3 in enriched conditioning, but to not overemphasize these findings we also state in both the results (line 238) and discussion (line 333) that other transcription factors and signalling pathways likely contribute to the phenotype and are also important. Please see our power analysis here below, where we calculated a difference in regeneration between groups from 10 to 20% of regenerating fibers, with a SD of 4 (control group) and 6 (experimental group), p value<0,05 and a power of 80%, resulting in a N:4 (orange box).

AEEC Animal Experimentation Sample Size

application.

Anticipated or Pilot Study Values			
	Mean*	n.	Dev #
Group 1	10	4	
Group 2	20	6	

Difference in mea 100 %

alpha level ("p" value)	Sample Size Needed in Each Group (n number)			
	95%	90%	80%	50%
0.05	7	5	4	2
0.02	8	7	5	3
0.01	9	8	6	3

INTERPRETATIONS

	A. Represents the minimum number of animals needed to attain statistical significance of $p < 0.05$ with an 80% probability
	B. Represents the optimum number of animals needed to attain statistical significance of $p < 0.05$ with a 90% probability
	C. Represents the maximum number of animals needed to attain statistical significance of $p < 0.01$ with a 95% probability

NOTES

The **minimum** number of animals used in any experiment should be $n=3$ group is $n=15$. If greater than this, a strong justification should be provided.
No replication of studies are permitted without strong justification.

* Mean differences can be estimated from previous studies or anticipated
 # SD can be estimated from pilot studies or previous publications of a

DEFINITIONS

The cells in the table above show the estimated number of subjects needed in each group in order to demonstrate a statistically significant difference at "p" values ranging from 0.05 - 0.01 and at varying levels of "power".

Power is the probability of finding a statistically significant difference at a

REVIEWERS' COMMENTS

Reviewer #1 (Remarks to the Author):

The additional data in Fig. 7H address my previous concern sufficiently. I look forward to seeing this in publication.

Reviewer #3 (Remarks to the Author):

I am happy with the responses to my comments